



# Adaptive clustering: A method to analyze dynamical similarity and to reduce redundancies in distributed (hydrological) modeling

Uwe Ehret[1], Rik van Pruijssen[1], Marina Bortoli[1], Ralf Loritz[1], Elnaz Azmi[2], Erwin Zehe[1]

[1]Institute of Water Resources and River Basin Management, Karlsruhe Institute of Technology - KIT, Karlsruhe, Germany
[2]Steinbuch Centre for Computing, Karlsruhe Institute of Technology - KIT, Karlsruhe, Germany

*Correspondence to*: Uwe Ehret (uwe.ehret@kit.edu)

**Abstract.** In this paper we propose adaptive clustering as a new way to analyse hydrological systems and to reduce computational efforts of distributed modelling, by dynamically identifying similar model elements, clustering them and inferring dynamics from just a few representatives per cluster. It is based on the observation that while hydrological systems

generally exhibit large spatial variability of their properties, requiring distributed approaches for analysis and modelling, there is also redundancy, i.e. there exist typical and recurrent combinations of properties, such that sub systems exist with similar properties, which will exhibit similar internal dynamics and produce similar output when in similar initial states and when exposed to similar forcing. Being dependent on all these factors, similarity is hence a dynamical rather than a static phenomenon, and it is not necessarily a function of spatial proximity.

We explain and demonstrate adaptive clustering at the example of a conceptual, yet realistic and distributed hydrological model, fit to the Attert basin in Luxembourg by multi-variate calibration. Based on normalized and binned transformations of model states and fluxes, we first calculated time series of Shannon information entropy to measure dynamical similarity (or redundancy) among sub systems. This revealed that indeed high redundancy exists, that its magnitude differs among variables, that it varies with time, and that for the Attert basin the spatial patterns of similarity are mainly controlled by

geology and precipitation. Based on these findings, we integrated adaptive clustering into the hydrological model. It constitutes a shell around the model hydrological process core and comprises: Clustering of model elements, choice of cluster representatives, mapping of results from representatives to recipients, comparison of clusterings over time to decide when re-clustering is advisable. Adaptive clustering, compared to a standard, full-resolution model run used as a virtual reality 'truth', reduced computation time to one fourth, when accepting a decrease of modelling quality, expressed as Nash-

Sutcliffe efficiency of sub catchment runoff, from 1 to 0.84.

We suggest that adaptive clustering is a promising tool for both system analysis, and for reducing computation times of distributed models, thus facilitating applications to larger systems and/or longer periods of time. We demonstrate the potential of adaptive clustering at the example of a hydrological system and model, but it should apply to a wide range of systems and models across the earth system sciences. Being dynamical, it goes beyond existing static methods used to

increase model performance, such as lumping, and it is compatible with existing dynamical methods such as adaptive time-



stepping or adaptive gridding. Unlike the latter, adaptive clustering does not require adjacency of the sub systems to be joined.

# 1 Introduction

Hydrological systems are often characterized by considerable spatial heterogeneity of relevant properties such as topography, soils, or vegetation (Schulz et al., 2006), and considerable dynamical variability due to time changing boundary conditions such as precipitation or radiation (Zehe and Sivapalan, 2009). If we are mainly interested in aggregated characteristics and dynamics of such systems, such as mean wetness, mean travel times, or discharge at a catchment outlet, lumped and conceptual modelling approaches such as Topmodel (Beven and Kirkby, 1979), HBV (Bergström, 1976) or hydrological response units (Flügel, 1996) will suffice. The merits of such models, easy set-up and short computation times, however come at a price: They conceptualize process patterns and redistribution processes and the underlying controls by means of effective dynamical laws, effective states, effective parameters and effective fluxes. This makes it difficult to use available observations for model parameterization and validation (Binley et al., 1989; Hundecha and Bárdossy, 2004; Kirchner, 2006) or to connect to other models.

Reductionist, physically based and distributed modeling approaches such as MIKE SHE (Abbott et al., 1986), HYDRUS (Šimunek et al., 1999) or CATFLOW (Zehe et al., 2001) on the other hand can be parameterized and validated by distributed observations, and provide physically meaningful, distributed answers based on distributed internal dynamics. The major drawbacks for their application are large demand of high-resolution data for model setup and operation, and a CPU demand that rapidly grows with system size. The latter problem can be dealt with by either crushing it with massive parallel computing (Kollet, 2010), or by reducing it by avoiding redundant computations. This can be done by exploiting patterns of similarity in time via adaptive time stepping (Minkoff and Kridler, 2006), in space by adaptive gridding (Pettway et al., 2010; Berger and Oliger, 1984), or both (Miller et al., 2006). Due to their generality, adaptive methods have been used to improve distributed modelling of a large variety of systems such as the universe (Teyssier, 2002), the atmosphere (Bacon et al., 2000; Aydogdu et al., 2019), oceans (Pain et al., 2005), and groundwater systems (Miller et al., 2006).

While these adaptive methods are highly useful, they all require direct adjacency in either time or space of the model elements to be joined. However, similarity, in both nature and models, is not necessarily a function of proximity only. Consider for example a landscape in the temperate mid-latitudes: It can typically be characterised by a relatively small number of recurrent combinations of properties, such as 'north-facing forested hillslopes with shallow soils underlain by schist', or 'plains used for agriculture, with deep soils underlain by marl'. The reason for observing only a limited number of such combinations lies in the long-term co-evolution of its constituents, adapting to the geological and climate setting and the related water, mass and energy flow regimes (Phillips, 2006; Dietrich and Perron, 2006; Troch et al., 2013; Schröder, 2006). While co-evolution thus reduces the potentially very large number of possible combinations, it does not imply that those parts of the landscape sharing the same property combinations are necessarily spatially connected. If we represent such



a landscape in a distributed model, we may therefore find many structurally similar (i.e. similar with respect to time-invariant properties) yet non-neighbouring model elements.

While structural similarity is a required condition for similarity, it is not sufficient: Only if two model elements share similar properties, are in similar initial states and are exposed to similar forcing, they will produce similar outputs based on similar internal dynamics (Zehe et al., 2014). To summarize: Landscape co-evolution limits the number of unique structural configurations in a landscape, thus providing a potentially large number of structurally similar elements in related models. Similarity is not a static but rather a dynamical property dependant on the interplay of structure, state and forcing, and similarity is not necessarily a function of spatial proximity.

Based on these premises, which are confirmed by the findings of Loritz et al. (2018), we suggest a method for adaptive clustering of model elements to avoid redundant computations. Adaptive clustering contributes to solving the computational challenges of hydrological modelling as formulated by Clark et al. (2017) and comprises several steps: Clustering of model elements, choice of cluster representatives, mapping of results from representatives to recipients, and comparison of clusterings over time to decide when re-clustering is advisable. We demonstrate it at the example of a conceptual, yet realistic and distributed hydrological model, fit to the Attert basin in Luxembourg using multi-variate calibration. Besides evaluating adaptive clustering in terms of computational gains and losses of modelling quality, we also discuss how the normalized and binned representations of model states used for clustering can be used for hydrological system analysis by revealing space-time patterns of similarity in the catchment.

The remainder of the manuscript is structured as follows: In section two, we explain the hydrological model and how it was set up for the Attert basin, introduce the main steps of adaptive clustering and describe its implementation in the hydrological model, and introduce the measures used to evaluate hydrological system similarity and adaptive clustering performance. In section three, we present and discuss results from the hydrological analysis of the Attert catchment. We also show the results from distributed modelling using adaptive clustering and compare it to a range of benchmark models with respect to computational efficiency and model quality. In section four, we summarize the results, draw conclusions and suggest further research.

## 2 Data and methods

### 2.1 The SHM hydrological model

Adaptive clustering, like adaptive gridding or adaptive time stepping, is a universal concept potentially applicable to many kinds of model types, reductionist or conceptual. One of the key requirements for adaptive clustering to make sense is the existence of similarity in natural systems, and, with it, similarity among the elements of distributed models representing them. For the physics-based, reductionist model CATFLOW (Zehe et al., 2001) applied to the Colpach catchment in the Attert basin, Loritz et al. (2018) already showed that among the 105 hillslopes used to represent the catchment, substantial and time dependent similarity occurred. However, CATFLOW's code structure does not allow a straightforward integration





and testing of adaptive clustering functionality. As uncomplicated implementation and testing of many variants was crucial during the development stage of adaptive clustering, we instead decided to employ a self-made, conceptual and distributed model that is numerically simple and offers full code control. This way we could focus on the challenge of implementing adaptive clustering as a generic feature of the model architecture, and on testing its feasibility for distributed computing. We named the model 'SHM' (Simple Hydrological Model) and it is largely based upon concepts known from established conceptual hydrological models like HBV (Bergström, 1976). We decided to set it up in the Attert basin in Luxembourg as for this catchment there exists a large store of hydrological knowledge we could build upon (Pfister et al., 2009; Juilleret et al., 2012), and we had access to a comprehensive data set compiled in the CAOS (Catchments as Organized Systems) project (Zehe et al., 2014).

### 2.1.1 SHM Structure

SHM is a distributed hydrological model, i.e. a catchment is divided into sub catchments which are typically a few square kilometres in size. The water stocks and fluxes in each sub catchment are represented in a conceptualized manner by a set of linked linear reservoirs (see Fig. 1). The choice of the type, number and linkage of reservoirs is based on the insights about the hydrological functioning of the Attert basin and suitable conceptualizations reported by Fenicia et al. (2014) and Fenicia et al. (2016). The model structural elements and all related equations are shown in Fig. 1 and Table 1. The first reservoir represents the unsaturated zone. Precipitation falling onto a sub catchment is divided into direct runoff and soil moisture replenishment as a nonlinear function of current soil moisture (the HBV beta store concept). Evapotranspiration draws water from the unsaturated zone storage. Direct runoff is split by a constant factor and replenishes two linear reservoirs, one representing interflow, the other representing base flow. Runoff from the interflow and base flow reservoirs are added and then enter the river system. The river system is represented by a linear reservoir cascade, where each element represents a river stretch of about one kilometre. The model is coded in Matlab and applies a straightforward non-iterative forward-in-time numerical scheme.



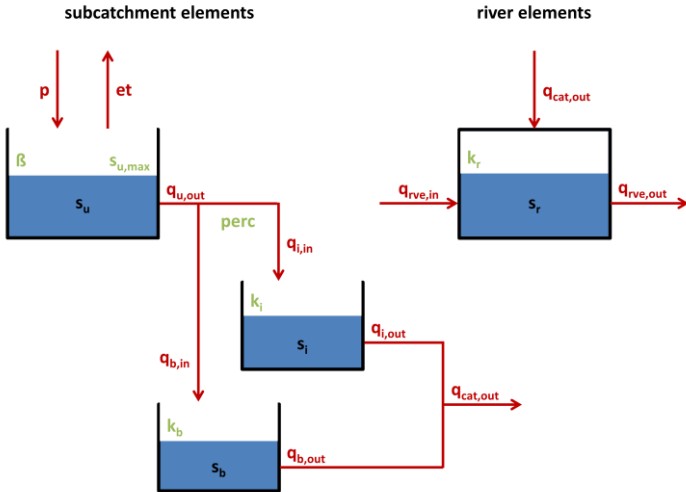

**Figure 1.** Structural elements, parameters (green), state variables (black) and fluxes (red) of the SHM model.





**Table 1.** SHM model equations and parameters

| Number | Equation[a] | Description |
|---|---|---|
| (1) | $vol\_p = p \cdot a$ | vol_p: precipitation volume [m³]<br>p: precipitation [m]<br>a: sub catchment area [m²] |
| (2) | $\Psi = \left(\dfrac{s_u}{s_{u,max}}\right)^{\beta}$ | $\Psi$: runoff coefficient [-]<br>$s_u$: storage in unsaturated zone reservoir [m]<br>$s_{u,max}$: maximum storage in unsaturated zone reservoir [m]<br>ß: shape coefficient [-] |
| (3) | $q_{u,out} = \Psi \cdot vol\_p$ | $q_{u,out}$: runoff from unsaturated zone reservoir [m³] |
| (4) | $et = et_{ref} \cdot k_c \cdot k_\theta$ | et: evapotranspiration [m³]<br>$et_{ref}$: reference evapotranspiration [m³][b]<br>$k_c$: crop coefficient [-][c]<br>$k_\theta$: soil moisture correction factor [-][d] |
| (5) | $q_{i,in} = q_{u,out} \cdot perc$ | $q_{i,in}$: inflow to interflow reservoir [m³]<br>perc: interflow-baseflow partitioning factor [-] |
| (6) | $q_{b,in} = q_{u,out} \cdot (1 - perc)$ | $q_{b,in}$: inflow to baseflow reservoir [m³] |
| (7) | $q_{i,out} = \dfrac{s_i}{k_i} \cdot a$ | $q_{i,out}$: runoff from interflow reservoir [m³]<br>$s_i$: storage in interflow reservoir [m]<br>$k_i$: interflow reservoir retention constant [-] |
| (8) | $q_{b,out} = \dfrac{s_b}{k_b} \cdot a$ | $q_{b,out}$: runoff from baseflow reservoir [m³]<br>$s_b$: storage in baseflow reservoir [m]<br>$k_b$: baseflow reservoir retention constant [-] |
| (9) | $q_{cat,out} = q_{i,out} + q_{b,out}$ | $q_{cat,out}$: runoff from sub catchment [m³] |
| (10) | $s_r = s_r + \sum\limits_i q_{rve,in,i} + \sum\limits_j q_{cat,out,j}$ | $s_r(t)$: storage in river element [m³]<br>$q_{rve,in,i}$: inflow from connected river element i [m³]<br>$q_{cat,out,j}$: inflow from connected sub catchment j [m³] |
| (11) | $q_{rve,out} = \dfrac{s_r}{k_r}$ | $q_{rve,out}$: runoff from river element [m³]<br>$k_r$: river element retention constant [-] |

[a]  in all equations, time subscripts t and t-1 are dropped for brevity

[b] evapotranspiration from reference surface (short grass) according to Penman (1956). Equations taken from DVWK (1996), section 5.3.1.

5  [c] crop coefficient as a function of land used and month of the year. Taken from Dunger (2006), Appendix 11. Value range [0.65, 1.3]



[d] adapted from Dunger (2006), section 4.5.8.4, Fig. 37 with the assumptions $k_\theta = 0$ for $s_u \leq 0$; $k_\theta = 1$ for $s_u \geq 0.8 * s_{u,max}$; $k_\theta = s_u / s_{u,max}$ for $0 < k_\theta < 0.8 * s_{u,max}$

### 2.1.2 SHM Attert

Our test site, the Attert basin, is located in the central western part of the Grand Duchy of Luxembourg and partially in
Eastern Belgium with a total catchment area of 288 km² up to gauge Useldange (Fig. 2). The landscape shows topographical,
geological and pedological diversity, with a small area underlain by sandstones in the South and Northeast, a wide area of
sandy marls in the centre part, and an elevated region underlain by schist in the North, which is part of the Ardennes massif.
The schist region reaches elevations up to 539 m a.s.l. and contains deeply incised river valleys. The Attert basin is situated
in the temperate oceanic climate zone, and snow-related processes play a negligible role. Precipitation is mainly associated
with westerly synoptic flow regimes and reaches annual amounts of about 850 mm (Pfister et al., 2005; Pfister et al., 2000).

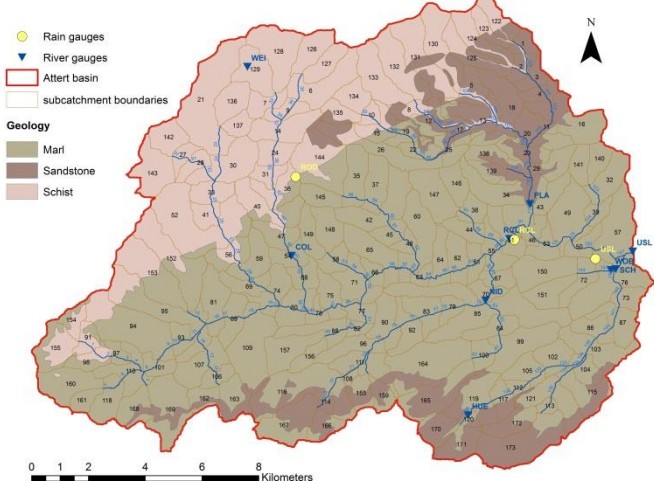

**Figure 2.** Map of the Attert basin up to gauge Useldange. Black labels are sub catchment ID's, blue labels are river element
IDs. Yellow and blue labels are rain and river gauge IDs, respectively. Detailed information about the gauges is given in
Table 2.

Setting up the SHM model to a catchment starts by GIS-based delineation of sub catchments and the river network using a
digital elevation model. For the Attert, a 5 m digital elevation model based on LIDAR scans provided by the Luxembourg
Institute of Science and Technology (LIST) was used. Each sub catchment was assigned a single land use, based on the
Corine Land Cover map provided by the European Environment Agency, and a single geology, based on the 'Carte
géologique détaillée 1:25000-1:50000' provided by the Luxembourg geological survey, by majority vote within each sub
catchment area. In the catchment, altogether five different land use classes and three geological classes occurred.



As in the Attert hydrological function is strongly controlled by geology (Fenicia et al., 2016), all soil-related model parameters (ß, $s_{u,max}$, perc, $k_i$, $k_b$) were kept the same among all sub catchments sharing the same geology and determined by calibration. Likewise, all parameters related to evapotranspiration ($k_c$, $k_\theta$) were kept the same among all sub catchments sharing the same land use class. These could, without calibration, be directly inferred from the land use (see Eq. 4 in Table 1). As we set all river elements to be approximately one kilometre in length, we assigned to all of them the same value for $k_r$, which we also determined by calibration.

Running the SHM model requires observed time series of precipitation, air temperature, air relative humidity, wind velocity and global radiation. For precipitation, data from three stations were available. While this is clearly not enough to represent the full spatial variability of precipitation, we deemed it sufficient to capture its main characteristics required for our conceptual distributed modelling approach. Each sub catchment was assigned precipitation from a single station using a nearest neighbour approach. As the rest of the required variables typically exhibit much less spatial variability than precipitation, we used observations from a single station each (see Table 2 and Fig. 2). All data were available in hourly resolution; we also executed the model in hourly time steps.

**Table 2.** Time series data used for model calibration and validation. All data were used in 1-h resolution for the period 2011/11/01 00:00 – 2016/10/31 23:00. Time reference is MEZ.

| ID | Full name | Data type and unit | Catchment area [km²] | Source |
|---|---|---|---|---|
| ROD_p | Roodt | precipitation [mm] | - | ASTA[a] |
| RCL_p | Reichlange | precipitation [mm] | - | AGE[b] |
| USL_p | Useldange | precipitation [mm] | - | ASTA |
| USL_ta | Useldange | air temperature at 2 m [°C] | - | ASTA |
| USL_rh | Useldange | air relative humidity at 2 m [%] | - | ASTA |
| ROD_v | Roodt | wind velocity at 3 m [m/s] | - | LIST[c] |
| MER_rg | Merl | global radiation [W/m²] | - | ASTA |
| COL_q | Colpach | discharge [m³/s] | 19.5 | LIST |
| WOB_q | Wollefsbach | discharge [m³/s] | 4.5 | LIST |
| PLA_q | Platen | discharge [m³/s] | 44.2 | LIST |
| USL_q | Useldange | discharge [m³/s] | 245 | LIST |

[a] Administration des services techniques de l'agriculture Luxembourg

[b] Administration de la gestion de l'eau Luxembourg

[c] Luxembourg Institute of Science and Technology





Adaptive clustering aims at enabling, at acceptable computation times, distributed modeling of relevant hydro-meteorological processes. Therefore a model used for its evaluation should not just perform well with respect to discharge at the catchment outlet, but it should provide reasonable distributed simulations of all relevant states and fluxes. To ensure this we applied a distributed multi-criteria calibration approach. For calibration we used data from the four-year period 2011/11/01 00:00 – 2015/10/31 23:00 and tested the model in the remaining one-year period 2015/11/01 00:00 – 2016/10/31 23:00. We started by joint calibration of all sub catchment parameters on catchment scale, i.e. against observed discharge at the catchment outlet Useldange, and against catchment-averaged observations of soil moisture and evapotranspiration. The unique set of available soil moisture data (observations from 18 sensors in the schist, 11 in the marls, and 19 in the sandstone region, all taken at 50 cm depth) was taken during the CAOS project. There is no direct representation of soil moisture in SHM, we therefore compared normalized and catchment-averaged observed soil moisture against normalized and catchment-averaged storage in the unsaturated zone ($s_u$). While this clearly did not allow quantitative conclusions, it was nevertheless very informative to compare the timing of relative minima and maxima and the shape of the overall dynamics. As direct observations of catchment-scale evapotranspiration rates were unfortunately not available, we used satellite-based estimates provided by EUMETSAT (Trigo et al., 2011) instead and compared them to catchment-averaged evapotranspiration rates of SHM. For each variable we measured model performance by Nash-Sutcliffe efficiency (Nash and Sutcliffe, 1970), and joined the three measures into a single objective function according to Eq. 12, with weights chosen to reflect the different quality of the observations.

$$NSE_{total} = 0.5 \cdot NSE_{discharge} + 0.3 \cdot NSE_{soil\ moisture} + 0.2 \cdot NSE_{evapotranspiration} \qquad (12)$$

After the first catchment-uniform and multi-criteria estimation of parameters, we refined the estimates of all soil-related model parameters by calibration against three gauges, each representative for a particular geology: Colpach for schist, Wollefsbach for marls, and Platen for sandstone. These parameters (see Table 3) were then assigned to all sub catchments sharing the same geology. After a few iterations of catchment-scale- and geology-specific calibration, we determined the final, distributed parameter sets as shown in Table 3. The main differences among geology-specific parameters appear for the retention behaviour of the interflow and the base flow reservoir ($k_i$ and $k_b$, respectively), which reflects the geology-specific hydrological functioning of the Attert basin as described by Fenicia et al. (2016): In the schist, dynamics is governed by a combination of two subsurface flow paths, in the marl, fast responses governed by near-surface flow paths prevail, while the sandstone areas are characterized by delayed responses governed by groundwater flow.

The catchment-scale performance measures for both the calibration and the validation period are shown in Table 4, and in Table 5 for performance at the gauges used for geology-specific and for catchment-wide calibration. The model achieves a catchment-scale, multi-objective Nash-Sutcliffe efficiency of 0.73 in the validation period; gauge- or criteria-specific efficiencies range from 0.61 for gauge Wollefsbach to 0.77 for gauge Useldange at the catchment outlet. For a visual comparison of observed and simulated discharge at Useldange in the year 2015, please see Fig. 5, panel b, lines 'observed'





and 'reference'. Overall we can conclude that the distributed SHM Attert hydrological model shows acceptable performance with respect to both internal state dynamics (represented by unsaturated zone storage) and the main fluxes leaving the system, evapotranspiration and discharge at the catchment outlet.

SHM Attert clearly still leaves room for improvement. However, we would like to stress that the model is not the central

5    topic of this paper. Rather, it serves as a test bed for the main topic, adaptive clustering, and we hold that it is fit for that purpose.

**Table 3.** Parameters of the SHM Attert found by calibration in the period 2011/11/01 00:00 – 2015/10/31 23:00. The parameters are described in Table 1.

| Geology | Target gauge ID | $s_{u,max}$ [m] | $\beta$ [-] | perc [-] | $k_i$ [-] | $k_b$ [-] | $k_r$ [-] |
|---------|-----------------|---------|-----|------|-----|-------|-----|
| Schist | COL_q | 0.17 | 5 | 0.5 | 44 | 500 | 1.1 |
| Marls | WOB_q | 0.17 | 3 | 0.7 | 20 | 3000 | 1.1 |
| Sandstone | PLA_q | 0.17 | 2 | 0.05 | 100 | 20000 | 1.1 |

**Table 4.** Catchment-scale performance measures (discharge: Nash-Sutcliffe efficiency at gauge Useldange, soil moisture and evapotranspiration: Nash-Sutcliffe efficiency of catchment averages) of the SHM Attert in the 5-year calibration period (2011/11/01 00:00 – 2015/10/31 23:00) and 1-year validation period (2015/11/01 00:00 – 2016/10/31 23:00). 'Combination' refers to the joint objective function according to Eq. 12.

| Series | Calibration | Validation |
|--------|-------------|------------|
| Discharge | 0.85 | 0.77 |
| Soil moisture | 0.80 | 0.66 |
| Evapotranspiration | 0.58 | 0.74 |
| Combination | 0.78 | 0.73 |

**Table 5.** Gauge-specific performance measures (Nash-Sutcliffe efficiency of discharge) of the SHM Attert in the calibration and validation period. Gauge locations are shown in Fig. 2, catchment sizes in Table 2.

| Geology | Gauge ID | Calibration | Validation |
|---------|----------|-------------|------------|
| Shist | COL_q | 0.78 | 0.65 |
| Marls | WOB_q | 0.66 | 0.61 |
| Sandstone | PLA_q | 0.79 | 0.74 |
| Catchment | USL_q | 0.85 | 0.77 |



## 2.2 Adaptive clustering

As explained in the introduction, the main goal of adaptive clustering is to reduce computational efforts of distributed and high-resolution modelling to facilitate application at larger scales or for longer periods of time. The main idea is to avoid redundant computations by clustering similar model elements, and then to infer the dynamics of all elements in a cluster from just a few representatives. For adaptive clustering to make sense, three preconditions must be fulfilled: Existence of i) many model elements of ii) the 'same kind', with potentially similar behaviour but iii) only weak interaction. If there were only few model elements, there would be nothing to cluster, if they would not be of the 'same kind' it would be impossible to assign results from representatives to represented elements, if there would be strong interaction, ignoring it, which is inevitable in adaptive clustering, would lead to large modelling error. These preconditions are largely fulfilled for sub catchments or hillslopes in distributed hydrological models: They occur in large numbers, there is only little or no interaction among them as they act in parallel, connecting to rivers, and as the critical zone in the landscape is composed of relatively few, typical, recurring combinations of its constituents (see the related discussion in the introduction and in Zehe et al., 2014), there is potential for similarity among model elements which we can exploit. It is important to note that even if two model elements were identical with respect to all of their time invariant (structural) properties, they could still behave differently when exposed to different forcing such as rainfall or solar radiation. Therefore, while similarity can have a strong time invariant component, and simple static clustering can be beneficial, the full potential of clustering will be exploited if it is treated dynamically (Loritz et al., 2018).

In the following two sections, we will first explain the main, application-independent steps of adaptive clustering, and then describe how we implemented it in the SHM model.

### 2.2.1 Main steps

The main steps of adaptive clustering are illustrated in Fig. 3, and we will explain the method along its sub plots. In the matrix shown in plot 'a', each row represents a single model element (sub catchment or grid element), and each column a time step. We start with a fully distributed model execution at the first time step, i.e. each sub catchment is executed separately to determine its own states and fluxes at time step $t_0$. Based on these fully distributed states and fluxes, similar sub catchments are combined in clusters (plot 'b'). Doing so involves two important choices: Choice of a suitable clustering algorithm and values of its hyper parameters, and choice of one or several sub catchment properties (structural properties, state or flux variables) by which the clustering is done and which we will in the following refer to as clustering control variable. Next, from each cluster a subset of its elements is selected, these serve as representatives for the entire cluster (plot 'c'). How these representatives are chosen, and how many of them, has a strong influence on the performance of adaptive clustering: A large number will lead to high modelling quality, but small computational gains, and vice versa. Generally, choosing the parameters controlling adaptive clustering is governed by the objective to maximize computational gains while





minimizing deterioration of modelling quality, compared to a fully distributed run. Once the representatives are found, the actual hydrological modelling for the next time step $t_1$ is done only for them (plot 'd'); the states and fluxes of all non-representative sub catchments (recipients) are taken over from the representatives using a suitable mapping technique (plot 'e'). At this step, the larger the difference between the computational costs of the clustering plus mapping and executing the

hydrological model is, the larger the benefit of clustering will be. Thus we expect a high benefit for numerically demanding models. However, the mapping comes at the risk of violating conservation laws, which are typically obeyed even in simple hydrological models. Also, there is no guarantee that a clustering done at some point in time, will still be valid at later times: Due to differences in forcing and structure, cluster members may not behave similarly all the time, and clusters may break apart, unite, or exchange elements over time. In such cases, maintaining an old clustering can result in inadequate mapping

and hence poor modelling results. To test this, a new clustering is done based on the values of the clustering control variable at time step $t_1$, but only for the representatives, as only they possess non-mapped values (plot 'f'). This clustering is compared to the clustering in effect of the representatives (plot 'g') using a suitable method. If the two clusterings are sufficiently similar (plots 'f'-'g'), the clustering in effect still holds and we can repeat steps (plots 'd'-'e') for the next time step $t_2$ (plots 'h'-'i'). As before, a new clustering is done based on the values of the clustering control variable for the representatives at time

step $t_2$ (plot 'j') and compared to the clustering in effect (plot 'k'). If this time the two clusterings are dissimilar (plots 'j'-'k'), it is an indication that the model internal patterns of similarity have shifted and a new clustering should be done. This should be done based on the values at $t_2$ of all sub catchments, and not just the representatives. However, for all recipients these are only mapped values inherited from the representatives, and a re-clustering based on these forced-to-be-similar values (due to the mapping) will be overly persistent. Of course, the best basis for clustering would be results from a fully distributed model

run, but performing such a run would bring to nought all previous computational savings of adaptive clustering. However, we found that running the model in full resolution even for a limited time prior to the point of re-clustering establishes a pattern of state and flux variability among the model elements close to that of a full resolution run. Choosing the length of that 'jump back in time' is again a trade-off between restoring the models' true variability pattern and additional computational expenses. In plot 'l', the jump goes back to $t_1$, and from there the model is run in full resolution, i.e. each sub

catchment forms its own single-element cluster (plot 'm'), until the time of re-clustering $t_2$ is reached again (plot 'n'). Based on the so-established close-to-full-resolution values of the clustering control variable of all sub catchments at $t_2$, a new clustering is done (plot 'o') and new representatives are selected (plot 'p'), which means we are back to the initial situation (plots 'a'-'c'), but have moved forward in time. These steps are then repeated until the end of the simulation period is reached.



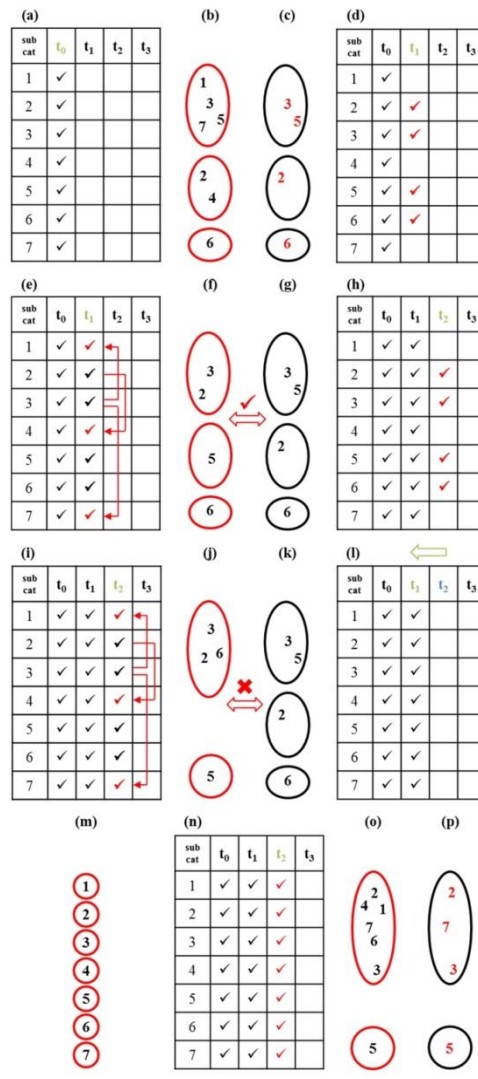

**Figure 3.** Main steps of adaptive clustering. A detailed explanation is given in the text. In each table, green letters indicate the current point in time; red colour indicates the action carried out.

### 2.2.2 Implementation in SHM Attert

5   Adaptive clustering is implemented as an optional module in SHM, and we tested it for the Atter catchment. At first we conducted a fully distributed model run, i.e. without any clustering, for the entire calibration and validation period. This is the best quality result the model can achieve, and serves as a virtual reality 'gold standard' against which all further runs applying adaptive clustering were compared. Next, we decided to use sub catchment runoff ($q_{cat,out}$, see Fig. 1 and Table 1) as a single clustering control variable for three reasons: Firstly, for catchment hydrologists runoff is the main variable of





interest, secondly, sub catchment runoff is influenced by all sub catchment states and fluxes, hence similarity of two sub catchments with respect to their runoff is a reasonable single-value indicator of overall similarity, thirdly we started with only a single control variable to keep things simple. For clustering and mapping we applied a straightforward yet effective approach based on binning: At first, we determined from the fully distributed run, separately for each sub catchment and

each variable, the quasi-observed ('quasi-observed' henceforth refers to results from the fully distributed virtual reality run) value range and used this to normalize each time series to a [0,1]-series. These dimensionless values were used for mapping from representatives to recipients: Recipients were forced to assume the representative's relative state (or flux), and these relative states (or fluxes) were then re-converted by each recipient's own value range to dimensionful values. If there was more than one representative in the cluster, we selected a single best one: The representative closest to the median value of

all representatives. The results of that single best representative were then mapped to its recipients. This method clearly leaves room for improvement, but we considered it good enough for a first proof-of-concept.

The normalized values were also used for clustering: We divided the [0-1] value range into 64 uniform bins, a number we deemed large enough to still resolve the non-binned values with sufficient detail, but small enough to ensure sufficiently populated clusters. All sub catchments with normalized values of the clustering control variable falling into the same bin

were assigned to the same cluster, i.e. any non-empty bin actually constitutes a cluster. With this approach, the number of clusters is limited to a minimum of one and a maximum of 64, and the number of clusters at a given point in time reflects the variability of the sub catchment's relative states at that time. Having established the clusters, their representatives were found by random picking controlled by three parameters (see Table 6): *Perc_reps* defines the total number of representatives, expressed as percentage of the total number of sub catchments in the model. Applied to each cluster, it provides a first

estimate about how many representatives should be picked from it. However, we found that besides controlling the total number of representatives, it was also useful to set a limit to the minimum and maximum number of representatives per cluster, which is controlled by parameters *min_reps_per_clus* and *max_reps_per_clus*.

The last but not the least part of adaptive clustering is to measure the degree of similarity between two clusterings, the clustering in effect which was established in the past, and the clustering based on current values, and to then decide whether

a re-clustering is required. In this context, it is important to note that it is not conducive to simply compare the members of equal bins, e.g. comparing the sub catchments in bin 12 of the clustering in effect with those in bin 12 of the clustering based on the current time step. Suppose for example that at the time step in the past, at which the clustering in effect was done, sub catchments one and two showed small runoff and were assigned to the lowest bin (= cluster one), while sub catchments three and four showed higher runoff and were assigned to the next higher bin (= cluster two). Suppose further that it then started

raining, uniformly increasing runoff from each sub catchment by the same rate, such that at a later time step, sub catchments one and two would be assigned to cluster two, and sub catchments three and four to the next cluster three. A direct comparison of clusters would indicate low similarity and thus suggest re-clustering, while in fact the grouping of the sub catchments remained the same, and they just moved together to new clusters with different labels. It was therefore necessary to first find a mapping between cluster labels of the two clusterings before measuring the similarity of their contents. For the





former we used the well-known Hungarian method (Kuhn, 1955; Munkres, 1957), for the latter we divided the number of elements in corresponding clusters by the total number of elements, which yields a degree of similarity between zero and one. The entire procedure is also illustrated in Fig. 4.

5  **Table 6.** Main parameters of adaptive clustering and values chosen for SHM Attert

| Name | Description | Value chosen |
|---|---|---|
| *num_bins* | The value range of a variable is divided into *num_bins* bins of uniform width | 64[a] |
| *perc_reps* | Ratio of sub catchments used as representatives and total number of sub catchments (soft constraint) | 10 |
| *min_reps_per_clus* | Minimum number of sub catchments to represent a cluster (hard constraint) | 1 |
| *max_reps_per_clus* | Maximum number of sub catchments to represent a cluster (hard constraint) | 8 |
| *sim_crit* | A re-clustering is triggered whenever the similarity between the clustering in effect and the clustering based on current data is < *sim_crit*. The value of *sim_crit* is the percentage of matching classifications, i.e. the number of main diagonal entries divided by the total number of entries in a confusion matrix | 55 |
| *sim_uncrit* | Controls the period for re-clustering: If a re-clustering is triggered, go back in time until the similarity between the initial and the current clustering is ≥ *sim_uncrit*. | 75 |

[a] for normalized variables with value range [0,1], this means bin edges [0, 0.0156, 0.0313, … , 0.9688, 0.9844, 1]





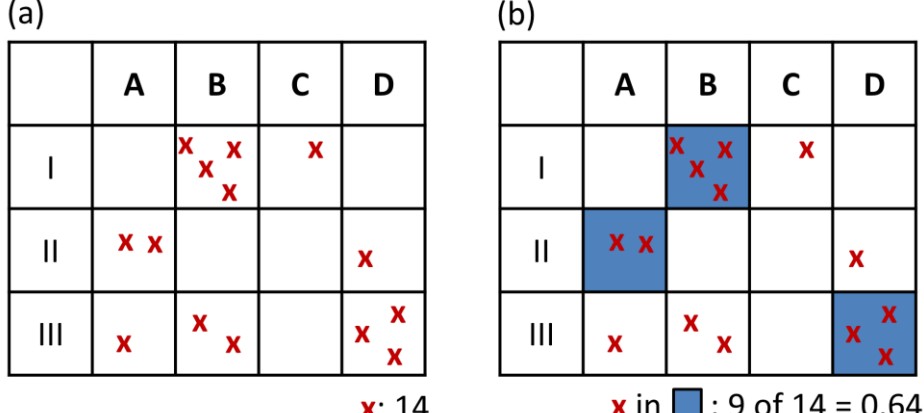

**Figure 4.** Evaluation of clustering similarity. (a): Clustering of elements (here: 14) according to two classifications, A-D and I-III. (b): Attribution of cluster labels between the two classifications, indicated by blue cells. Cluster label I corresponds to cluster label B, II to A, III to D, and C has no match. The similarity of the two clusterings (here: 0.64) is measured by the number of elements in corresponding clusters (elements in the blue cells, here: 9) divided by the total number of elements (14).

With this approach, we calculated the similarity between the clustering in effect and the clustering based on the values of the current time step. If they disagreed substantially, i.e. similarity fell below the acceptance limit set by *sim_crit* in Table 6, it was time for a re-clustering. In such a case, a jump back in time was triggered. The length of the jump is controlled by parameter *sim_uncrit*, which like *sim_crit* is a similarity threshold: The jump goes back to the last time this threshold was still exceeded. Depending on the prevailing hydro-meteorological situation, this can be shorter or longer, but it will never extend beyond the point of the last re-clustering. From that point in the past, the model was run, in full distribution, up to the point in time when the jump back was triggered. Back at this point, with the diversity of the sub catchment states now close to that of a full-resolution run, a re-clustering is done (see plots 'l'-'o' in Fig. 3 and related explanations in the text).

The values of all parameters for the SHM Attert application are shown in Table 6, and were found by manual iterative trial-and-error with the objective of maximizing computational savings while minimizing quality loss.

## 2.3 Design of experiments and evaluation

As explained in the introduction, adaptive clustering can be applied for two distinct but related questions: Analysis of dynamical patterns of similarity within hydrological systems (Loritz et al., 2018), and reducing computational efforts by exploiting them. While the first question is interesting in its own right, addressing the second involves addressing the first, as optimal computational savings can only be achieved if the nature of the patterns is understood. We therefore addressed both questions. As explained in section 2.2, we first conducted a fully distributed model run for the combined calibration and validation period, which served as the basis for the similarity analysis, and as a virtual reality 'gold standard' to compare adaptive clustering runs against.



### 2.3.1 Hydrological system analysis

How to measure the dynamical similarity among sub catchments? They differ in size, and many of their states and fluxes are a function of size, therefore a direct comparison of values is not useful. Also, for comparability it would be desirable to apply the same method of comparison irrespective of the variable considered, and results to be in the same units.

All of these requirements are fulfilled when we consider the [0,1]-normalized and binned time series of states and fluxes as described in section 2.2 rather than the original ones, and apply the methods and measures of information theory to them. At each point in time, the occupations of the 64 bins together form a histogram, which can be normalized to a discrete probability distribution by dividing with the total number of sub catchments populating the bins. The overall variability (or similarity, or redundancy) of the sub catchments for any state or flux of interest can then be measured in unit bit by the

Shannon entropy H of the corresponding discrete probability distribution according to Eq. 13. This approach was also used by Loritz et al. (2018) for hydrological system analysis; a more detailed introduction to concepts, measures and applications of information theory is given in Neuper and Ehret (2019), Singh (2013), and Cover and Thomas (1991).

$$H(X) = -\sum_{x \in X} p(x) \ log_2 \ p(x) \qquad (13)$$

We can then produce time series of entropy for any state or flux to reveal time-patterns of similarity, and we can calculate their time-averages as an overall measure of similarity, which we can use to compare different states and fluxes. It is an interesting property of Shannon entropy H, which for a discrete distribution with a given number of bins, there exists an upper and a lower bound: If all elements fall into a single bin, its probability hence being one, and that of all other bins being zero, H will take its minimum value, zero. Applied to our sub catchments this would indicate that they are all in the same

relative state, and similarity is at its maximum. At the other extreme, maximum dissimilarity is represented by a uniform distribution, which is also referred to as maximum entropy distribution, whose entropy $H = log_2(n)$, where n is the number of bins. As we used the same 64 bins for all variables, the same upper bound of 6 bit applies to all variables.

### 2.3.2 Adaptive clustering

With respect to adaptive clustering, two aspects are important: Computational savings, and quality losses. The savings we

measured in two ways: Firstly by simply measuring the time of model execution, while keeping all but the adaptive clustering settings constant: All runs were performed on the same machine and with no additional processes running in parallel. We verified the reproducibility of the timings by repeating selected runs several times; the observed spread of results was very small. Timing measures the entire modeling effort, i.e. both for running the hydrological model, and for the overhead of adaptive clustering caused by similarity analysis, clustering, clustering comparison, etc. While this is a

straightforward and effective way to evaluate and compare different adaptive clustering settings for a given hydrological model, its drawback is that the relative contributions from the hydrological model and from the overhead cannot be



distinguished. Conceptual as it is, the effort associated with executing hydrological processes of the SHM Attert is small, and the overhead of adaptive clustering can therefore quickly exceed the gains of reducing the number of sub catchments to execute. For hydrological models with more elaborate physical process representations such as MIKE SHE (Abbott et al., 1986), CATFLOW (Zehe et al., 2001) or HydroGeoSphere (Brunner and Simmons, 2011; Davison et al., 2018) however, the

relative costs of the overhead may be much smaller, making the computational expenses of adaptive clustering pay off much faster. We therefore evaluated the modeling effort in a second way, by counting at each time step the number of sub catchments for whom hydrological processes were actually executed. If a time step was visited twice, once in the forward and once in the jump back mode, we added the two countings. Multiplying these counts with a characteristic execution time for a single sub catchment and the total number of sub catchments in a hydrological model yields an estimate of total

computational effort if the share of the overhead is comparably small.

We decided to use Nash-Sutcliffe efficiency of sub catchment runoff $q_{cat,out}$ as a measure of modeling quality for the same reasons we decided to use it as clustering control variable: it is a hydrologically meaningful, comprehensive and single-valued measure. Using the quasi-observed time-series of the fully distributed run as a reference, we calculated the NSEs for each sub catchment, and then a catchment-wide NSE by calculating their area-weighed mean. While this is an aggregated

quality measure, it is more informative than calculating the NSE of discharge at the basin outlet, as it avoids possible error compensation during discharge convolution in the river network.

We conducted many adaptive clustering runs using different parameter settings, and we compared the results in terms of quality and effort to several benchmark cases: The 'reference' case consisted of running SHM without any adaptive clustering functionality, i.e. fully distributed and without any adaptive clustering overhead. The 'static' case consisted of running SHM,

with the adaptive clustering functionality implemented but its parameters set such that there was only one initial clustering, where each sub catchment was put in a separate cluster, and any jump backs were suppressed. This means that adaptive clustering was in action, causing its computational overhead, but nevertheless the model ran in the same fully distributed manner as the standard case. We also established a 'static optimal' case based on an offline, prior similarity analysis of the sub catchments: As we deliberately set the SHM Attert up in a straightforward manner, based on a limited number of

observables with only limited variability, to keep the focus on the most important controls of similarity, there was a considerable number of sub catchments agreeing in all of their structural and functional properties except size (see Table 7), meaning that by definition their normalized states and fluxes had to be identical throughout the entire simulation period. The 173 sub catchments could thus be grouped into only 24 time-invariant yet optimal clusters. 'Optimal' here means that there is no within-cluster variability, and any single sub catchment picked from a cluster is a perfect representative of all others. This

is of course a simplified and idealized case due to the simplified set up of the model. If further structural properties and forcing had been used for model setup and operation, and in higher resolution, very likely such a drastic reduction would not have occurred. Nevertheless we used this 'static optimal' case to evaluate the merits of advance knowledge about static sub catchment similarity. The model setup and operation was equal to the static case, but this time the static optimal clustering was used instead of the 'each sub catchment in a single cluster' setting.

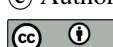



**Table 7.** Sub catchments of SHM Attert, grouped into 24 static clusters by agreement in attributes geology, land use and meteorological forcing. Sub catchment locations are shown in Fig. 2. The possible number of unique attribute combinations is 3 * 5 * 3 = 45. The cluster size range is [1, 28], the average number of elements in a cluster is 7.2.

| Geology class[a] | Land use class[b] | Rain gauge[c] | # of cluster members | Sub catchment IDs |
|---|---|---|---|---|
| 1 | 1 | ROD | 13 | 6, 7, 8, 10, 14, 15, 40, 127, 131, 133, 134, 135, 137 |
| 1 | 1 | RCL | 1 | 124 |
| 1 | 2 | ROD | 7 | 21, 36, 91, 142, 144, 152, 153 |
| 1 | 2 | RCL | 1 | 123 |
| 1 | 3 | ROD | 13 | 9, 27, 30, 33, 41, 52, 126, 128, 129, 136, 143, 154, 155 |
| 1 | 3 | RCL | 1 | 122 |
| 1 | 4 | ROD | 5 | 24, 28, 31, 130, 132 |
| 2 | 1 | ROD | 22 | 19, 22, 26, 47, 71, 74, 82, 88, 94, 95, 96, 97, 98, 106, 108, 110, 116, 118, 157, 161, 168, 169 |
| 2 | 1 | RCL | 28 | 29, 34, 38, 44, 48, 51, 53, 62, 64, 67, 70, 79, 83, 84, 85, 90, 92, 99, 111, 117, 119, 138, 139, 146, 147, 150, 158, 164 |
| 2 | 1 | USL | 15 | 32, 72, 73, 76, 86, 102, 103, 104, 105, 112, 113, 115, 121, 140, 151 |
| 2 | 2 | ROD | 19 | 35, 42, 45, 54, 56, 58, 59, 65, 68, 69, 75, 78, 80, 81, 89, 93, 101, 145, 148 |
| 2 | 2 | RCL | 6 | 16, 43, 46, 60, 61, 100 |
| 2 | 2 | USL | 6 | 39, 49, 50, 57, 87, 141 |
| 2 | 3 | ROD | 2 | 149, 167 |
| 2 | 4 | ROD | 7 | 37, 77, 107, 109, 156, 160, 162 |
| 2 | 4 | RCL | 5 | 25, 55, 63, 120, 159 |
| 2 | 5 | ROD | 2 | 66, 163 |
| 3 | 1 | RCL | 6 | 2, 4, 11, 12, 23, 125 |
| 3 | 2 | ROD | 1 | 114 |
| 3 | 2 | RCL | 1 | 20 |
| 3 | 3 | ROD | 1 | 166 |
| 3 | 3 | RCL | 5 | 1, 3, 5, 13, 18 |
| 3 | 4 | RCL | 4 | 17, 165, 170, 171 |
| 3 | 4 | USL | 2 | 172, 173 |

5    [a] 1: Schist, 2: Marl, 3: Sandstone



[b] 1: Meadow, 2: Agriculture, 3: Coniferous forest, 4: Broad-leaf forest, 5: Sealed area

[c] ROD = Roodt, RCL = Reichlange, USL = Useldange (compare Table 2)

## 3 Results and discussion

### 3.1 Hydrological system analysis

As described in section 2.3.1, we will discuss hydrological similarity with a focus on three aspects: Dynamical behavior, time-averaged entropies, and spatial patterns. For the first, time series of Shannon entropy for selected variables of the SHM Attert model (fully resolved run) are shown in Fig. 5, panel a. For better visibility, the plot is restricted to a single year. High values indicate high variability across sub catchments, low values high redundancy. First of all, it is striking that for all

variables, entropies remain well below the benchmark maximum entropy shown in red, and are often close to zero. Obviously there is a lot of redundancy among the sub catchments, which can be exploited by clustering. The second important observation is that entropies change with time. While the rate of change differs among variables (e.g. high for interflow storage $s_i$, and low for base flow storage $s_b$), it is present for all of them, which emphasizes that clustering should be done dynamically rather than statically. The entropy of our clustering control variable, $q_{cat,out}$, shows a high correlation

with discharge magnitude as shown in panel b: In times of rising and high discharge, entropies are high, which is likely due to the interplay of spatially distributed precipitation and catchment states, and the onset of fast runoff components, which may differ among sub catchments. As fast runoff components are typically dormant through times of recession, and precipitation is uniformly zero for all sub catchments, low flow is accompanied by low entropies. All of these observations agree well with the findings of Loritz et al. (2018).





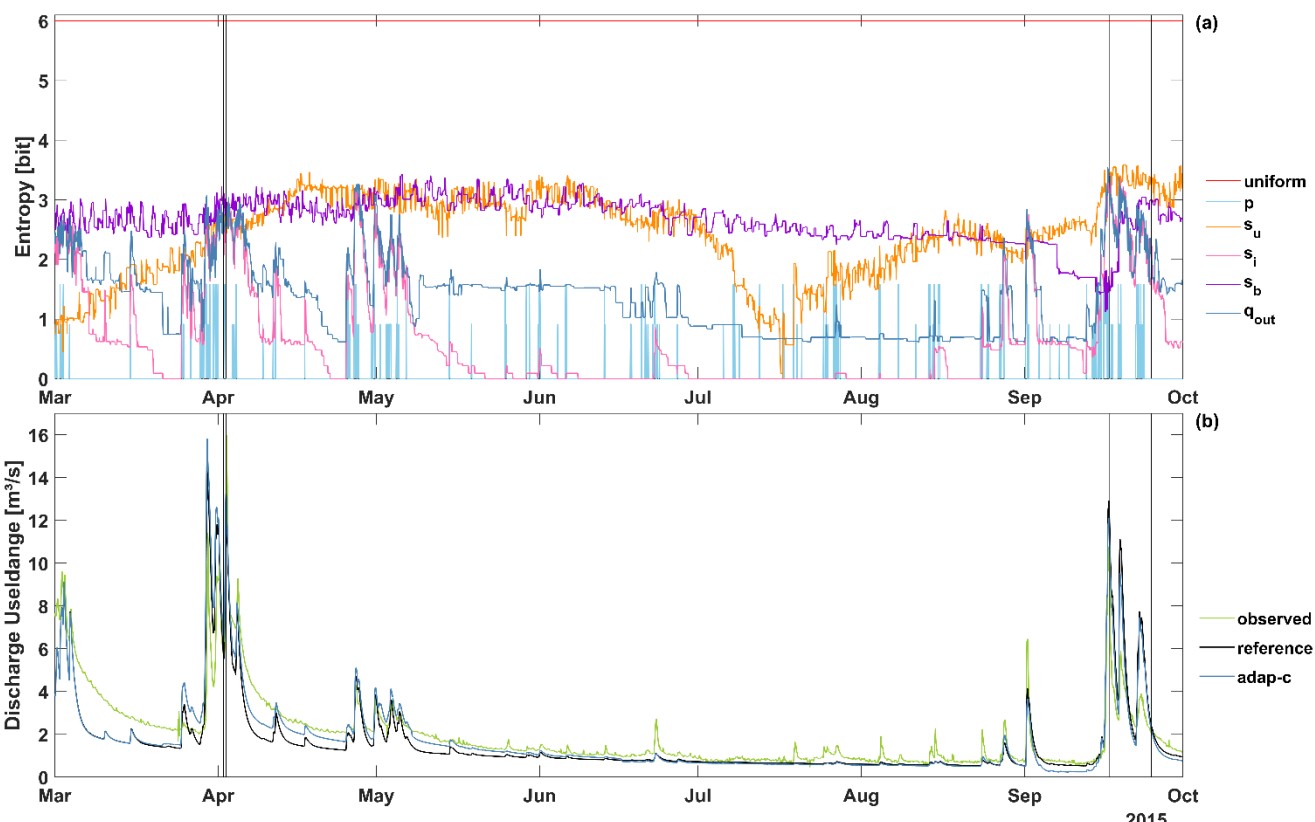

**Figure 5**. Panel a: Time series of Shannon entropy of distributions of normalized and binned sub catchment states. Distributions are based on the fully distributed model run. 'P' is precipitation, '$s_u$' is unsaturated zone storage, '$s_i$' and '$s_b$' are interflow and base flow storage, '$q_{out}$' is sub catchment runoff (see Fig. 1 and Table 1). 'Uniform' indicates the benchmark maximum entropy of 6 bit for a 64-bin distribution. Black vertical lines indicate times of special interest for which spatial maps are shown in Fig. 6. Panel b: Discharge time series at

catchment outlet gauge Useldange. 'Observed' are real-world observations, 'reference' are virtual reality quasi-observations from the fully distributed model run, 'adap-c' are from an adaptive clustering run with optimized parameters as shown in Table 6.

Time-averaged entropies for all SHM variables are shown in Table 8. As already indicated in Fig. 5, the values differ quite substantially among the variables, with precipitation 'p' showing the lowest entropy of only 0.1 bit, and base flow related

variables $s_b$ and $q_{b,out}$ showing the highest entropy of 2.88 bit. The low value of precipitation entropy can be explained by two effects: Firstly, during the frequent times of no rain, precipitation entropy is also zero as all stations show the same value, and secondly even if it rains, at most three different bins of the distribution can be occupied as we measure precipitation by only three stations and assign these values unchanged to the sub catchments by nearest neighbor mapping. This limits entropy to a maximum of merely $\log_2(3) = 1.58$ bit. The high values for base flow can be explained by the strong, geology-

induced differences of the base flow behavior across the catchment (compare the strongly differing values of $k_b$ in Table 3),





and the fact that due to the slow-changing nature of base flow, these differences prevail for a long time, keeping entropies high throughout the year (see Fig. 5, panel a). Interestingly, the entropies of several variables are identical ($q_{u,out}$, $q_{i,in}$, and $q_{b,in}$; $s_i$ and $q_{i,out}$; $s_b$ and $q_{b,out}$). This is not a coincidence, but can be explained by how they are related: $q_{i,in}$, and $q_{b,in}$ are percentages of $q_{u,out}$; runoff from both the interflow and the base flow reservoir are linear functions of only the respective storages (see Fig. 1 and Table 1). All of these relations are entropy-preserving transformations, i.e. the entropies of all variables involved are necessarily equal.

**Table 8.** Mean entropy of all normalized and binned SHM Attert states and fluxes of the reference run, calculated as time-average of entropies from all time steps in the 6-year test period (2011/11/01 00:00 – 2016/10/31 23:00). The states and fluxes are explained in Table 1.

| State or flux | Entropy H [bit] |
|:---:|:---:|
| uniform | 6[a] |
| p | 0.10 |
| et | 0.76 |
| $s_u$ | 2.38 |
| $q_{u,out}$ | 0.18 |
| $q_{i,in}$ | 0.18 |
| $s_i$ | 1.23 |
| $q_{i,out}$ | 1.23 |
| $q_{b,in}$ | 0.18 |
| $s_b$ | 2.88 |
| $q_{b,out}$ | 2.88 |
| $q_{cat,out}$ | 1.84 |

[a] Entropy of the benchmark uniform distribution

In Fig. 6 we show spatial patterns of normalized and binned values of the clustering control variable $q_{cat,out}$ for selected points in time. Plots in the left column ('a'-'e') are based on the reference run, our virtual reality, and we will focus on these in the following. We selected the times such as to cover a wide range of different hydrological situations (compare the black vertical lines in Fig. 5). Plots 'a' and 'b' are both in spring and related to the same rainfall-runoff event, the last in a sequence of three, with plot 'a' showing the values just before the onset of precipitation, and plot 'b' at the time of peak runoff. Comparing the plots it is obvious that the general magnitude of runoff has increased, indicated by the main colours shifting from red (low values) to yellow (intermediate values). While this was to be expected, additionally we can see that the spatial





pattern of similarity also shifted from a geology-dominated pattern, reflecting the geology-based parametrization of sub catchments, to a pattern reflecting the joint influence of both geology and the spatial distribution of rainfall (see geological map and rain gauge locations in Fig. 2). Interestingly, while the grouping of the sub catchments into clusters obviously changed, the overall number of clusters did only increase by two, from 12 to 14. The next plot, 'c', shows a very different

situation at the end of a long summer drought: Most sub catchments show very low runoff, only the sand stone areas, where groundwater flow dominates, maintain runoff above their absolute minimum. Overall, the entire catchment is in a very homogeneous state and sub catchments group into only three clusters. This uniform state comes to a sudden end with the onset of precipitation (plot 'd'), increasing the diversity of sub catchment runoff to 18 clusters and the development of a spatial pattern which is mainly influenced by rainfall spatial distribution and only to a lesser degree by geology: the sand

stone area is not as clearly separated from the other geologies as usual. Finally, after a period of extended rainfall (plot 'e'), a spatial pattern similar to the initial one in plot 'a' has re-established, but overall runoff magnitudes are still lower; a heritage of the long dry summer.

Altogether, the analysis of dynamical behavior, time-averages, and spatial patterns revealed that similarity of (normalized and binned) sub catchment states and fluxes is generally high, but also that the degree of similarity and its controls strongly

vary with time. In the next section we will discuss if, to which degree, and at which price adaptive clustering can capitalize on this.





**reference run**  **adaptive clustering run**



**Figure 6**. Spatial maps of normalized and binned values of the clustering control variable, $q_{cat,out}$, for all sub catchments and for selected points in time as marked by the black vertical lines in Fig. 5. Colors indicate the values, which correspond to the bin numbers ranging from



(lowest normalized state, red) to 64 (highest normalized state, blue). Values in the left column are from the fully distributed reference run, values in the right column are from an adaptive clustering run with optimized parameters as shown in Table 6.

## 3.2 Adaptive clustering

As explained in section 2.3.2, we evaluated adaptive clustering with respect to both computational effort and associated
quality losses against several benchmark cases. Fig. 7 shows the results as a two-dimensional plot. The black square indicates the standard case of a model run in full resolution and without any adaptive clustering functionality, which took 816 seconds. As this is our virtual reality 'truth', the model shows perfect performance indicated by an NSE of 1. Then we integrated the adaptive clustering functionality into the model, but by choice of its parameters enforced a fully distributed run. This 'static' run is indicated by a black triangle. As to be expected, the model still performed perfectly, but the overhead
of adaptive clustering increased computation times by 707 seconds to a total of 1523 seconds! This is almost a doubling compared to the standard case, a computational extra cost which clustering needs to over-compensate in order to be worth the effort. And indeed it does, even for the simple case of static optimal grouping (red triangle) as shown in Table 7. Representing 173 sub catchments by 24 representatives (one per cluster) reduced computation time to 233 seconds, despite the overhead, at no loss of quality. How does adaptive clustering compare to that? The blue dots depict results for various
parameter choices (we tested many more but show only the pareto-optimal results), revealing a general pattern of trade-off between effort and quality: The more computational effort we accept, the higher the modelling quality, and vice versa.

The red dot represents the, in our eyes, optimal tradeoff based on the optimized parameter set shown in Table 6. The related computation time is 207 seconds, NSE is 0.84. This means that compared to the standard case, we have reduced computation time to one-fourth at the price of worsening NSE by 0.16. What this means in terms of discharge at the catchment outlet is
shown in Fig. 5, panel b: While differences between the fully distributed ('reference') and the adaptive clustering ('adap-c') run are visible, they are generally much lower than the differences between the former and the real observation ('observed'), which also had a lower overall NSE of 0.77 (Table 5). This is encouraging, but compared to the only slightly higher computation time, and no quality loss, of the static optimal clustering we may ask whether a dynamical treatment is worth the effort. For the given model, and if computational efficiency is the main concern, our answer would be 'No'. However,
not always will we be able to group model elements into such a small and static number of clusters (Table 7), and with zero within-cluster variability, as for SHM Attert. In fact, we deliberately designed it to be an only minimally adequate representation of true catchment variability, and for many other models the number of static clusters may be close to the total number of model elements, and adaptive clustering hence may be the better option.





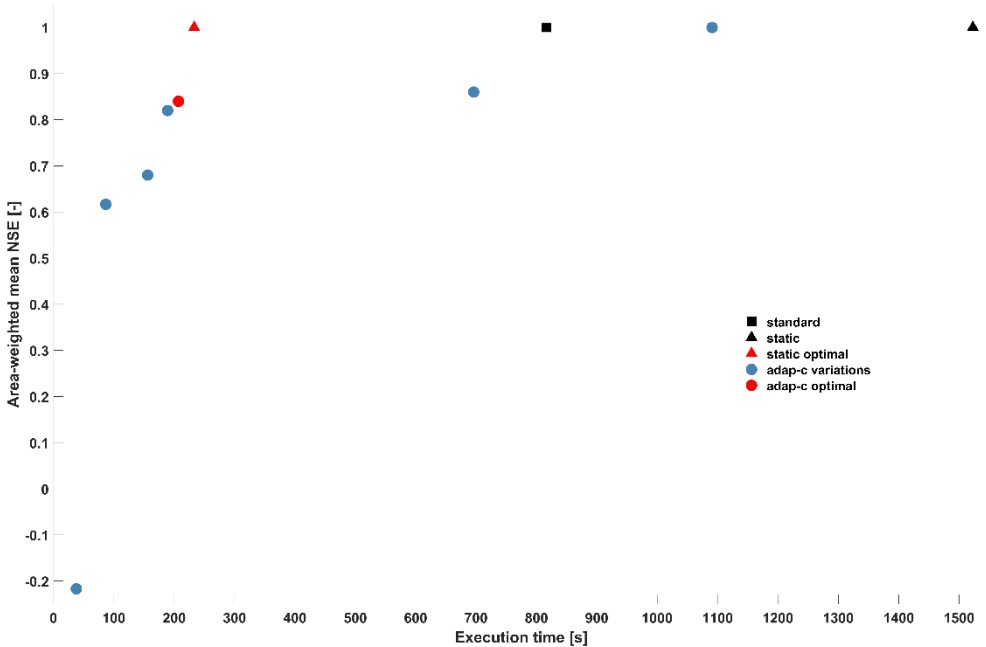

**Figure 7**. Performance of model runs with respect to effort, measured by execution time, and quality, measured by Nash-Sutcliffe efficiency of sub catchment runoff $q_{cat,out}$ (area-weighted mean of all sub catchments, and $q_{cat,out}$ of the full-resolution run as the reference). 'Standard': Full resolution, no adaptive clustering overhead; 'static': Full resolution, but with adaptive clustering overhead; 'static optimal': Time-invariant optimal clustering, with clusters shown in Table 7; 'adap-c variations': adaptive clustering with various parameter settings; 'adap-c optimal': optimal adaptive clustering with parameters shown in Table 6.

Fig. 8 allows a closer look at the behavior of adaptive clustering, based on the 'adap-c optimal' model run. Panel a shows, again for the year 2015 only, similarity between the currently used (but determined in the past) clustering and the clustering based on current values. Similarity strongly varies with time, and each time it falls below the critical threshold given by *sim_crit*, the model jumps back to the last time of acceptable similarity given by *sim_uncrit*, and then does a fully distributed run forward, to when the jump back was triggered. While the jump back periods help establishing a close-to-fully distributed model state for optimal re-clustering, the associated modeling effort is high, as can be seen in panel b. However, it is also apparent that jump backs occur only occasionally, and there may be periods of low dynamics, like July-September 2015, where a given clustering remains appropriate for a long time. During the entire six-year simulation period, only 165 re-clusterings were required, i.e. on average one every eleven days.




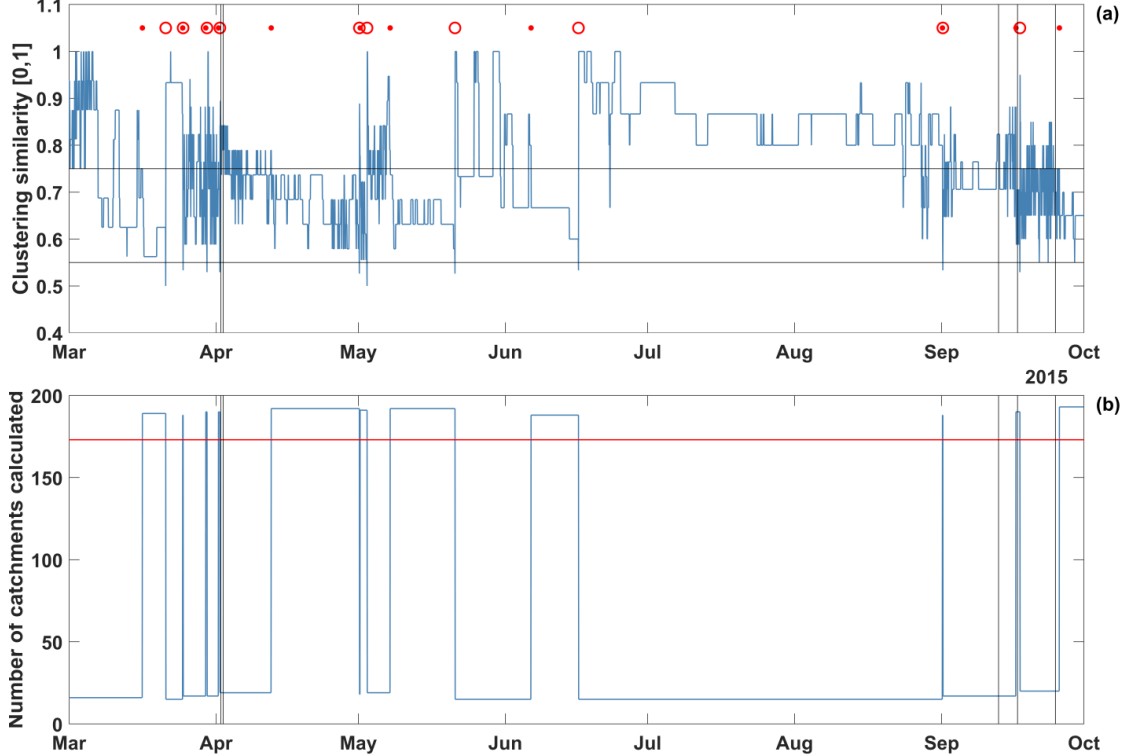

**Figure 8**. Panel a: Time series of clustering similarity, based on the 'adap-c optimal' model run. Black vertical lines indicate times of special interest (compare Fig. 5 and Fig. 6), black horizontal lines indicate the upper and lower similarity thresholds for re-clustering (see Table 6). Each red circle indicates when a re-clustering was triggered, and for each the next red dot to the left indicates the end of the related jump back in time. Panel b: Number of sub catchments per time step for which hydrological processes were calculated. The red horizontal line indicates the total number of sub catchments (173) in SHM Attert.

Despite adaptive clustering restricting hydrological process execution to the representatives only, it maintained the general spatial patterns of the fully distributed run. This can be seen when comparing corresponding plots in the left and the right column of Fig. 6. The main pattern characteristics (geology as the main control, strong variation of mean values and pattern variability over time) are also apparent in the adaptive clustering plots, even if at times there is no full agreement: For example, comparing plots 'b' and 'g' reveals a smaller influence of the rainfall pattern in the adaptive clustering case, for plots 'd' and 'i' the opposite is true. Generally, as to be expected, the overall richness of the patterns is reduced in the adaptive clustering case, i.e. the number of clusters is always smaller, but nevertheless the temporal variation of the number of clusters is similar.



## 4 Summary and conclusions

In this paper we proposed, described and test-applied adaptive clustering as a new way to reduce computational efforts of distributed modelling, while largely maintaining modelling quality. This is done by identifying, in a dynamical manner, similar model elements, clustering them and inferring distributed dynamics from just a few representatives per cluster.

We started from the observation that hydrological systems generally exhibit spatial variability of their properties, and that this variability is non-negligible if distributed dynamics are of interest, which then requires distributed modelling. We further hypothesized that despite this variability, there is also redundancy, i.e. there exist typical and recurrent combinations of properties, meaning that many model elements exist with similar properties, which will exhibit similar internal dynamics and produce similar output when in similar initial states and when exposed to similar forcing. Similarity is hence not a static but

rather a dynamical property dependant on the interplay of these factors, and similarity is also not necessarily a function of spatial proximity.

Based on these premises we developed the adaptive clustering method, and demonstrated it at the example of a conceptual, yet realistic and distributed hydrological model (SHM), fit to the Attert basin in Luxembourg by multi-variate calibration. Adaptive clustering comprises several steps: Clustering of model elements, choice of cluster representatives, mapping of

results from representatives to recipients, and comparison of clusterings over time to decide when re-clustering is required. We explained these steps in general, and its implementation in the SHM Attert model in particular. We used normalized and binned transformations of model states for both clustering and for measuring overall variability (or redundancy) via Shannon information entropy of the resulting discrete probability distributions. Analysing time series of entropy of model states and fluxes revealed that indeed high redundancy among model elements exists, that the degree of redundancy varies with time,

and that the spatial patterns of similarity are mainly controlled by geology and precipitation. We then evaluated adaptive clustering with respect to both computational gains and losses of model quality against several benchmark models. Compared to a standard, full-resolution model run used as a virtual reality 'truth', computation time could be reduced to one fourth, when accepting a decrease of modelling quality, expressed as Nash-Sutcliffe efficiency of runoff, from 1 to 0.84. Re-clustering occurred at irregular intervals mainly associated with the onset of precipitation, but on average the patterns of

similarity were quite stable, such that during the entire six-year simulation period, only 165 re-clusterings were carried out, i.e. on average once every eleven days.

Our tests and analyses were conducted in the virtual reality of a fully distributed model run, due to a lack of equally comprehensive observations. However, due to the good overall agreement of the model and the available distributed and multi-variate observations, we are confident that our main conclusion, namely that adaptive clustering is a promising tool for

both hydrological system analysis and for accelerating distributed hydrological modelling, also holds with respect to the real world. We suggest that adaptive clustering can, in addition to existing methods of exploiting dynamical similarities, such as adaptive gridding or adaptive time-stepping, help improving distributed modelling of dynamical systems. A limitation of the method lies in the potential violation of conservation laws when mapping results from representatives to recipients.



What's ahead? For the study we selected sub catchment runoff as the single variable for both clustering control and model evaluation. This was mainly based on hydrological reasoning, and clearly we should test other and/or additional variables for clustering control, and evaluate adaptive clustering in terms of its effect on all of the model's state and flux variables. At last, we have tested adaptive clustering at the example of a relatively simple, conceptual hydrological model with limited internal

variability. However, the potential savings of adaptive clustering will increase with the level of hydrological process detail in the model, but on the other hand difficulties of clustering will increase with model internal variability. It will therefore be interesting to test adaptive clustering in more advanced models, hydrological or other, such as MIKE SHE (Abbott et al., 1986), HydroGeoSphere (Brunner and Simmons, 2011; Davison et al., 2018), Noah-MP LSM (Niu et al., 2011), or the Community Land Model CLM (Lawrence et al., 2019),  where computation times are indeed a challenge.

*Data availability.* The precipitation data of stations Roodt and Useldange, the air temperature, relative humidity and global radiation data are publicly available from the Administration des services techniques de l'agriculture Luxembourg ASTA at http://www.agrimeteo.lu/  (last  access: 2019/08/20). The precipitation and discharge data at station Reichlange are available upon request from the Administration de la gestion de l'eau Luxembourg AGE at https://www.inondations.lu/ (last access:

2019/08/20). All other discharge data and the wind velocity data are available upon request from Luxembourg Institute of Science and Technology LIST at https://www.list.lu/ (last access: 2019/08/20). The EUMETSAT-based LSA-SAF evapotranspiration products are publicly available from http://landsaf.ipma.pt (last access: 2019/08/20). The soil moisture data are available upon request from Theresa Blume (blume@gfz-potsdam.de) and Markus Weiler (markus.weiler@hydrology.uni-freiburg.de). The digital elevation model is available upon request from LIST. The 2012

Corine Land Cover data are publicly available from the Copernicus sites of the European Environment Agency EEA at http://land.copernicus.eu/pan-european/corine-land-cover/clc-2012/view (last access: 2019/08/20). The geological maps are available upon request from the Service géologique, Administration des ponts et chaussées Luxembourg at http://www.geologie.lu/geolwiki/index.php/Cartes_g%C3%A9ologiques (last access: 2019/08/20). The SHM Attert including the adaptive clustering functionality and all code used to conduct the analyses in this paper are publicly available

at https://github.com/KIT-HYD/SHM-Attert-Adaptive-Clustering.

*Author contributions.* UE coded the SHM model and the adaptive clustering algorithm and wrote the manuscript. UE, RvP, RL, EA and EZ developed the adaptive clustering concept together. RvP set up the SHM Attert model, MB calibrated and validated it.

*Acknowledgements.* We gratefully acknowledge support by Deutsche Forschungsgemeinschaft DFG and Open Access Publishing Fund of Karlsruhe Institute of Technology (KIT). This research contributes to the "Catchments As Organized Systems" (CAOS) research group funded by the Deutsche Forschungsgemeinschaft DFG.



We acknowledge the providers of the hydro-meteorological data used in this study: The Administration des services techniques de l'agriculture Luxembourg (ASTA), the Administration de la gestion de l'eau Luxembourg (AGE), the Luxembourg Institute of Science and Technology (LIST), the EUMETSAT LSA-SAF consortium, and the CAOS research unit, especially Theresa Blume and Markus Weiler for providing the soil moisture data. We further acknowledge the
providers of the spatial data used in this study: Service géologique, Administration des ponts et chaussées Luxembourg, and LIST.

*Competing interests.* The authors declare that they have no conflict of interest.

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
