# Peer review of "Adaptive clustering: Reducing the computational costs of distributed (hydrological) modeling by exploiting time-variable similarity of model elements"

_Hydrology and Earth System Sciences, 2020_

## Referee Comment (RC1) · Shervan Gharari (Referee) · 19 Mar 2020

The manuscript introduces an adaptive clustering for grouping the hydrological elements of a distributed model. I enjoyed reading the manuscript and I think the work is interesting and eventually can be published. However, I should raise few overall points mostly regarding the presentation and the aim of the study.

1- The use of English language is far from being perfect. It took me more than usual to read through the manuscript due to insufficient use of English language. The sentences are very long, the wording are sometimes very awkward. For example, the first sentence of the abstract is very hard to follow. It is amalgamation of information which,

at the end, does not say much about the intention of this study (the intention being representative model with lower computational demand). Here I give examples from the text. Page 4, line 4, "offers full code control"; full code control is very subjective. Page 4, line 21, "…straightforward non-iterative forward-in-time numerical scheme". Page 7, line 19 "majority vote". Page 11, line 6 "for adaptive clustering to make sense". Page 13, line7 "gold standard" instead of synthetic. Page 14, line 3, "keep things simple". I would strongly recommend the authors to have this manuscript proofread by a native speaker.

2- The Introduction seems to be superficial. I would say the paper is about representation of the system in a model vs computational time/resource. In land [surface] modeling community there is significant body of literature devoted to the effect of grid size (computational burden vs spatial representation) and example of them can be Melsen et al., 2016 (and many more). This is the case in hydrological rainfall/runoff models as well (Liu et al., 2016 and many more). In its current form the Introduction starts with general reflection on sophisticated processed-based models; then moves to concept of co-evolution (which is not directly relevant to the message this study wants to convey) and then comes down to clustering. I would suggest to re-organize the Introduction to reflect on pervious works on computational burden vs spatial representation, clustering and its application in hydrological similarities and finally make it clear what the reader should expect from this paper.

3- Section 2 can be better organized. There is still mention of the CATFLOW in this section. I would suggest the authors to shorten the text in section 2.1 and directly explain the model. I suggest using transpiration coefficient instead of crop coefficient in Table-1 as the region is partly covered with forest. It seems to me that the routing is only a linear reservoir (eqation-11). Can this representation simulation lag function or unit hydrograph which is often used in the models? I would say no, therefore the highest streamflow peaks are the same as precipitation peaks. The are kr and kb identifiable/related? it seems to a redundancy in the two processes/parameters. It is

unclear how the computational units are set up, based on superposition of all the geospatial data or at the sub-basin level (later in Table-7 it becomes more apparent that the setup is at sub-basin level). Which data is directly used in setting up the model? I would suggest providing a table or bullet points for that; now it is very scattered around. I suggest removing equation 12, to a separate Section with entropy as measure of performance and uncertainty. Why there is NS values report in this Section? is the satellite based evaporation a result of more sophisticated model (such as a land model)? Section 2.2 is the heart of this manuscript. I would say it should be presented separately in a Section before the model is presented as this approach is model independent. Don't give sub-section "main steps" title (2.2.1). why section 2.3.1 is called "hydrological system analysis"; it is about entropy as a measure of uncertainty (as NS is a measure of performance). Section 2.3.2 is again called adaptive clustering, similar to section 2.2. and again, in Section 2.3.2 the authors are referring to CATFLOW and MIKE SHE, etc.

4- Section 3 is also rather hard to follow as well. The primary message of the manuscript is about tradeoff between spatial representation and computational resources (time). It is best to start from Figure-6 and 7 and then move to Figure-5 for example. Page 20 line 9, why it is "striking" that the entropies are lower than the uniform? I would always expect so. It is also expected that the entropy is lower for the recession and higher for rising discharge. This is kind of similar to the heteroscedasticity assumption on the error as well (more diffused with higher discharges). If only observation is used with varying error assumption, higher streamflow will have higher entropy and lower streamflow will have lower entropy. I don't really see the information the first and second paragraph of Section 3.1 provide. The third paragraphs, starting with "In Fig.6", is also kind of obvious. Are these what we have trained the model to do? The result is like a self-fulfilling prophecy (I will elaborate on this later).

5- The final Section is lacking proper discussion on what we have read in this manuscript. What can be the take home message for a modeler who want to model this basin in the future. A bullet point conclusion of this study is also appreciated.

[Figure]

6- For me personally, moving from the world of conceptual models to land models, I would like to question the motivation of this study. Although saving time is valuable but having method that needs model re-run or updating for more complex model is terribly cumbersome. This is the reason why the authors have chosen to use SHM rather than CATFLOW for example. I also didn't really understand the model set up, are we looking at comparison of models with clustering with a synthetic case? If that is the case, then the comparison of what is the best representation in a specific time of the year is dependent on the most elaborative set up and its simulations (basically the most computationally expensive model should be set up and simulated). Also, one might say there are easier time stepping approaches for lower computational costs. For example, for the high flow the model temporal resolution can be maximum (1 day) and during recessions it can be up to couple of days. These type of approach to reduce computational time is much easier to implement. Moreover, if computational time is considered, I would say Matlab is not the best choice. Land models implemented in C or Fortran can handle hourly simulation (with much more variables, IO traffic) much faster. BTW, I also missed the modeling time stepping; please clarify. I would suggest the authors to clarify their methodology of the clustering. Please make more time and maybe better visualization to convey the message here. As I said earlier, I would suggest the author to allocate a full Section for the clustering method they have proposed.

Overall, the manuscript is interesting but I think much more work is needed to meet the publication standard. I will suggest Major revision for this manuscript. I hope my comments can help the authors better present their good work.

With kind regards,

Shervan Gharari

Liu, H., Tolson, B.A., Craig, J.R. and Shafii, M., 2016. A priori discretization error metrics for distributed hydrologic modeling applications. Journal of hydrology, 543,

pp.873-891.

Melsen, L., Teuling, A., Torfs, P., Zappa, M., Mizukami, N., Clark, M. and Uijlenhoet, R., 2016. Representation of spatial and temporal variability in large-domain hydrological models: case study for a mesoscale pre-Alpine basin. Hydrology and Earth System Sciences, 20(6), p.2207.

---

## Referee Comment (RC2) · Anonymous Referee #2 · 27 Mar 2020

The main objective of this paper was to propose a new way to analyze hydrological systems using adaptive clustering. The study is interesting by dynamically identifying and clustered similar model elements. Representatives per cluster inferred the dynamics. Although the application of the proposed framework looks promising, additional investigations and explanations are necessary before this paper can be published in HESS. In the next sections, I outline my major comments and suggestions that should allow the authors to improve their manuscript. Thus, my recommendation is to be accepted after the following points are clarified: 1. The dynamic, comparability, and similarity were emphasized in the proposed framework for adaptive clustering. However, the definitions of dynamic, comparability, and similarity are ambiguous. For example, the

dynamic in hydrological simulation includes temporal dynamic and spatial dynamics. In particular, time scales for the application of hydrological models in the temporal dynamics are critical. 2. The main result of high redundancy in geology and climate information is obvious. The key reason is the intricately linear or nonlinear correlation among the subsystem. The technique of adaptive clustering extracts the nonlinear correlation but not nonlinear information. Further analysis of the nonlinear relationship between subsystems is suggested. 3. Only one basin, Attert basin, was studied. Moreover, Figure 2 and the relevant analysis did not elaborate on the geology and climate conditions, which are vital for the clustering of subsystems and hydrological simulation. The number of study areas is suggested to increase. The generalization of the framework proposed in this study is recommended to be demonstrated. 4. One critical issue is that the definitions of some technical terms were vague, such as aggregated characteristics and dynamics of such systems, co-evolution, catchment-uniform, and multi-criteria estimation. Please explain them in detail or replace them with easy-to-understand terms to enhance the readability of the study. 5. Unfortunately, the grammatical errors, confusing sentences, redundant vocabulary, and an erratic writing style hinder the message that the authors want to convey, and in some cases, render some statements ambiguous or even mistaken. I recommend that the authors encourage further to undergo a resubmission process. Data and methods section and results section are confusing, vague wording. I suggest elaborating on the description of the adaptive clustering. Otherwise, it is hard to understand how does analyze dynamical similarity. The main steps in 2.2.1 section are suggested to describe in points. 6. The resolution of Figure 2 is low. The information cannot be extracted. The explanation for Figure 3 and Figure 4 is difficult to understand. 7. In the data and methods section, excessive writing space is used for introducing the SHM model and its structure, which are not vital in this study. The structure of this article is suggested to adjust and enhance readability. In the results and discussion section, the principal results and conclusions are suggested to summarize briefly. 8. How to estimate the weights in Eq. 12? I think weight has a significant influence on the streamflow simulation in different

phases, which is essential for the applicability of the proposed framework. Moreover, the Nash-Sutcliffe efficiency only prefers the simulation accuracy of high flow. 9. How to identify and estimate the representatives which have a strong influence on the performance of adaptive clustering. 10. Importantly, the mechanism for the improvement of model performance was not discussed. For example, What operations lead to improvements in model performance (also involving high flow, middle flow, or low flow)?

---

## Author Comment (AC1) · 6 Apr 2020

**Responses to comments posted by Referee #1**

We thank the first referee, Shervan Gharari, for reviewing our article and providing feedback. The referee comments identify some unclear issues and help to improve the presentation of our research. In the following, we answer to all of the comments one by one. The Referee comments are in blue. Please note that our enumeration differs from that of the referee, because we split several comments into individual points. The original order of the referee comments however remains unchanged.

Comment 1: 1- The use of English language is far from being perfect. It took me more than usual to read through the manuscript due to insufficient use of English language. The sentences are very long, the wording are sometimes very awkward. For example, the first sentence of the abstract is very hard to follow. It is amalgamation of information which, at the end, does not say much about the intention of this study (the intention being representative model with lower computational demand). Here I give examples from the text. Page 4, line 4, "offers full code control"; full code control is very subjective. Page 4, line 21, ". . .straightforward non-iterative forward-in-time numerical scheme". Page 7, line 19 "majority vote". Page 11, line 6 "for adaptive clustering to make sense". Page 13, line7 "gold standard" instead of synthetic. Page 14, line 3, "keep things simple". I would strongly recommend the authors to have this manuscript proofread by a native speaker.

Reply 1: We will re-read the paper and re-write with shorter sentences and streamlined vocabulary where necessary. Also, in the production phase, the manuscript will undergo further copy-editing.

Comment 2: 2- The Introduction seems to be superficial. I would say the paper is about representation of the system in a model vs computational time/resource. In land [surface] modeling community there is significant body of literature devoted to the effect of grid size (computational burden vs spatial representation) and example of them can be Melsen et al., 2016 (and many more). This is the case in hydrological rainfall/runoff models as well (Liu et al., 2016 and many more). In its current form the Introduction starts with general reflection on sophisticated processed-based models; then moves to concept of co-evolution (which is not directly relevant to the message this study wants to convey) and then comes down to clustering. I would suggest to re-organize the Introduction to reflect on pervious works on computational burden vs spatial representation, clustering and its application in hydrological similarities and finally make it clear what the reader should expect from this paper.

Reply 2: We thank the referee for pointing at useful literature related to the effect of grid size in land surface modeling. We will integrate it in a revised version of the manuscript. However we do not agree that the introduction is superficial. We discuss conceptual modeling and its main shortcomings to show the merits of distributed modeling. We then discuss the computational

challenges of distributed modeling and how they are mitigated by adaptive time stepping and adaptive gridding. We discuss the shortcomings of available adaptive gridding methods, namely that they require spatial adjacency, and argue that similarity of sub systems of natural systems is not necessarily limited to neighboring elements. Taken together, this shows that adaptive clustering i) can be useful, ii) it has novel aspects compared to existing methods of adaptive gridding, and iii) similarity is not only an artificial effect caused by representing natural systems in models, but it is also property of the real-world system (although the inevitable simplifications associated with representing real-world system in models can increase similarities). We therefore prefer to keep the structure of the introduction as it is.

Comment 3: 3- Section 2 can be better organized. There is still mention of the CATFLOW in this section. I would suggest the authors to shorten the text in section 2.1 and directly explain the model.

Reply 3: Thank you for this suggestion, which was also made by referee #2. During manuscript preparation, it was also a matter of discussion among the authors whether the SHM model description should be at such a prominent place, as it just serves as a testbed for the demonstration of adaptive clustering. Nevertheless, as SHM has not been described elsewhere so far, and as knowledge of the model structure and parameterization is important to understand its behavior in terms of dynamical similarity, we think the model should still be presented in detail. We will move most of section 2.1 (The SHM hydrological model) to the Appendix. In the main text, we will give a very brief introduction to the model and will refer to the Appendix.

Comment 4: I suggest using transpiration coefficient instead of crop coefficient in Table-1 as the region is partly covered with forest.

Reply 4: The correction factors we use to adjust Penman reference ET from the reference surface (short grass) are available in Dunger (2006) not only for crops, but for a range of vegetation cover, including coniferous forest and deciduous forest. So while the term 'crop coefficient' is widely used for this kind of correction factor, it is not limited to crops (see e.g. http://www.fao.org/3/X0490E/x0490e0b.htm#crop%20coefficients), but we agree that the term is misleading. In a revised version of the manuscript, we will replace 'crop coefficient' by 'vegetation correction factor'.

Comment 5: It seems to me that the routing is only a linear reservoir (eqation-11). Can this representation simulation lag function or unit hydrograph which is often used in the models? I would say no, therefore the highest streamflow peaks are the same as precipitation peaks.

Reply 5: Thank you for raising this point. A single river element is indeed represented by a single linear reservoir. There are altogether 147 such river elements in the model, which together form a linear reservoir cascade. A linear reservoir cascade can reproduce both translation and

retention effects associated with streamflow. Therefore, streamflow peaks in the model do not necessarily coincide with the precipitation peaks. To avoid misunderstandings, we will in a revised version of the manuscript mention the 147 river elements and the linear reservoir cascade.

Comment 6: The are kr and kb identifiable/related? it seems to a redundancy in the two processes/parameters.

Reply 6: Both parameters are indeed retention constants of linear reservoirs, however they strongly differ in magnitude. Kr is the retention constant of a single river element. As all river elements in the model are of about the same length (1 km), we used the same kr for all river elements. Its value of 1.1 hours was found by maximizing the agreement between observed and simulated streamflow in river stretches where up- and downstream gauges were available (this way we could see the effect of translation and retention in the river stretch).

Kb is the retention constant of the base flow reservoir. We determined kb values, separately for each geology (see p.9, lines 4-28), by maximizing agreement of simulated and observed streamflow during times of summer low flow. Kb values are in the range of 500 hours (for Schist) and 20000 hours (for Sandstone), i.e. at least two orders of magnitude higher than kr. We are therefore confident that there is little redundancy between these processes/parameters.

Comment 7: It is unclear how the computational units are set up, based on superposition of all the geospatial data or at the sub-basin level (later in Table-7 it becomes more apparent that the setup is at sub-basin level). Which data is directly used in setting up the model? I would suggest providing a table or bullet points for that; now it is very scattered around.

Reply 7: Indeed the computational units are the sub-basins (termed sub catchments in the manuscript). The setup of the sub catchments, and how land use and geology are assigned to each of them is described on p.7, line 15 – p.8, line 14. We agree with the referee that it would be helpful to show Table 7 (currently in section 2.3.2 'Adaptive clustering') already in section 2.1.2 ('SHM Attert'). The reason for placing it in section 2.3.2 is that it shows the static optimal clustering of sub catchments, which is an important benchmark for adaptive clustering. As we will move most of section 2.1 to the Appendix (please see our reply to comment 3), we decided to leave the table in section 2.3.2 also in a revised version of the manuscript, but will refer to it in the 'SHM Attert' section. Also, to make clearer which rain gauge data were used for each sub catchment, we will add to Figure 2 lines separating the area of influence of each rain gauge (please see our reply to referee 2, comment 4).

Comment 8: I suggest removing equation 12, to a separate Section with entropy as measure of performance and uncertainty. Why there is NS values report in this Section?

Reply 8: The main topic of this manuscript is to introduce and show a proof-of-concept of adaptive clustering. SHM Attert in this context is a means to an end rather than a central topic. We therefore decided to discuss all aspects of SHM set up, calibration and validation in short form and in a single section, and to keep it separate from the discussion of entropy related to system analysis and evaluation of adaptive clustering. We therefore prefer to keep the discussion of calibration/validation results and entropy separate, all the more because we will move most of section 2.1 (The SHM hydrological model) to the Appendix (see our reply to comment 3).

Comment 9: is the satellite based evaporation a result of more sophisticated model (such as a land model)?

Reply 9: Yes, the ET estimates are produced by forcing a SVAT model (a simplified version of the ECMWF TESSEL SVAT scheme) by Land-SAF radiation products (DSSF, DSLF and AL) and ECMWF meteorology. A detailed description is given in Trigo et al. (2011), section 3.1, and on the LSA-SAF pages (https://landsaf.ipma.pt/en/products/evapotranspiration-energy-flxs/met/).

Comment 10: Section 2.2 is the heart of this manuscript. I would say it should be presented separately in a Section before the model is presented as this approach is model independent.

Reply 10: We agree. We will move most of section 2.1 to the Appendix to give more visibility to the main topic of the manuscript (please see our reply to comment 3).

Comment 11: Don't give sub-section "main steps" title (2.2.1).

Reply 11: Agreed. In a revised version of the manuscript, we will use a different title.

Comment 12: why section 2.3.1 is called "hydrological system analysis"; it is about entropy as a measure of uncertainty (as NS is a measure of performance).

Reply 12: Actually the section is about the question of how to analyze hydrological systems in terms of the time-varying similarity of its sub-elements (see p. 17, lines 2-4), and we suggest entropy as a suitable measure to do so. However, we agree that the section title does not perfectly reflect that. In a revised version of the manuscript, we will instead use 'Entropy as a measure of hydrological similarity'.

Comment 13: Section 2.3.2 is again called adaptive clustering, similar to section 2.2. and again, in Section 2.3.2 the authors are referring to CATFLOW and MIKE SHE, etc.

Reply 13: Respectfully, we do not understand the concerns of the referee here.

Comment 14: 4- Section 3 is also rather hard to follow as well. The primary message of the manuscript is about tradeoff between spatial representation and computational resources (time). It is best to start from Figure-6 and 7 and then move to Figure-5 for example.

Reply 14: We agree that the primary message of the paper is about how adaptive clustering can reduce computational efforts of distributed modeling while maintaining, by and large, modelling quality. However, adaptive clustering is built on the observation that i) hydrological similarity among sub systems exists, and ii) that this similarity is time-variant. If i) would not be true, there would be no potential for clustering; if ii) would not be true, time-invariant clustering would do the job. We therefore think it is important to first show to the reader results with respect to i) and ii), which we do in section 3.1, before moving on to the main topic in section 3.2. However, the referee comment reveals that we can do better to show the connection between these topics. In a revised version of the manuscript, we will add a related explanation at the beginning of section 3.2.

Comment 15: Page 20 line 9, why it is "striking" that the entropies are lower than the uniform? I would always expect so. It is also expected that the entropy is lower for the recession and higher for rising discharge. This is kind of similar to the heteroscedasticity assumption on the error as well (more diffused with higher discharges). If only observation is used with varying error assumption, higher streamflow will have higher entropy and lower streamflow will have lower entropy.

Reply 15: We agree that it is no surprise to see entropies below the entropy of the corresponding uniform distribution. 'Striking' here refers to the fact that entropies are well below the uniform entropy, and often close to zero. In our opinion this is indeed noteworthy, and it shows the high potential for adaptive clustering. In a revised version of the manuscript, we will make this point clear. We also agree with the referee that hydrologists have known since long that the degree of similarity between sub systems varies with the hydrological situation. The point we want to make here is that i) entropy of normalized, binned distributions of states and fluxes expresses this in a conveniently dimensionless way, and ii) that we make use of this knowledge.

Comment 16: I don't really see the information the first and second paragraph of Section 3.1 provide.

Reply 16: These sections show several important points: i) hydrological similarity among sub systems exists, and ii) this similarity is time-variant. iii) the temporal and spatial patterns of similarity are in accordance with hydrological expertise, which increases confidence that our way of expressing similarity in terms of entropies is reasonable. In a revised version of the manuscript, we will summarize these points at the section end.

Comment 17: The third paragraphs, starting with "In Fig.6", is also kind of obvious. Are these what we have trained the model to do? The result is like a self-fulfilling prophecy (I will elaborate on this later).

Reply 17: Please see or reply to comment 16.

Comment 18: 5- The final Section is lacking proper discussion on what we have read in this manuscript. What can be the take home message for a modeler who want to model this basin in the future. A bullet point conclusion of this study is also appreciated.

Reply 18: Respectfully we disagree. The point of the paper is not about how to model the Attert basin in the future, it is about the potential of adaptive clustering (as the referee correctly states in comment 14). We will however check in both the abstract and the conclusions if we can convey this message better.

Comment 19: 6- For me personally, moving from the world of conceptual models to land models, I would like to question the motivation of this study. Although saving time is valuable but having method that needs model re-run or updating for more complex model is terribly cumbersome. This is the reason why the authors have chosen to use SHM rather than CATFLOW for example.

Reply 19: The referee correctly states one main motivation for this study: saving computation time. This is already a more than sufficient reason, as the referee will surely agree, as he also works on concepts to make high-resolution land surface modeling more efficient (Gharari et al, 2020). In addition, the concept of adaptive clustering yields valuable insights in the time-and space patterns of similarity among sub systems, which, we daresay, is a useful contribution to hydrology research. We have chosen SHM for the proof-of-concept as any hydrologist can easily connect to it, and hence we can show the effects of adaptive clustering in an environment familiar to most hydrologists. We agree with the referee that implementing adaptive clustering in more advanced models will be more demanding, but also the gains will be higher (see p. 29, lines 5-9). So it will be well worth the try.

Comment 20: I also didn't really understand the model set up, are we looking at comparison of models with clustering with a synthetic case? If that is the case, then the comparison of what is the best representation in a specific time of the year is dependent on the most elaborative set up and its simulations (basically the most computationally expensive model should be set up and simulated).

Reply 20: Yes, the benchmark for evaluating the model performance with adaptive clustering is the full-resolution model run (i.e. without adaptive clustering), which is our ground truth (see p.16, lines 20-22). As we have shown that the model correctly reproduces various observations (p. 10, lines 1-6), we are confident that this virtual reality approach is valid.

Comment 21: Also, one might say there are easier time stepping approaches for lower computational costs. For example, for the high flow the model temporal resolution can be maximum (1 day) and during recessions it can be up to couple of days. These type of approach to reduce computational time is much easier to implement.

Reply 21: The model time stepping is 1 hour (not 1 day) and was kept the same throughout all simulations (please see our reply to comment 23). We agree with the referee that adaptive time stepping methods can dramatically reduce computation times, without substantial losses of simulation quality. We mention this on p. 2, line 20. The point is that adaptive clustering can be used in addition to adaptive time stepping. In that sense, acknowledging the merits of adaptive time stepping does not render adaptive clustering redundant.

Comment 22: Moreover, if computational time is considered, I would say Matlab is not the best choice. Land models implemented in C or Fortran can handle hourly simulation (with much more variables, IO traffic) much faster.

Reply 22: We agree that Matlab is not the best choice for high-performance computing. But this is not the point of our manuscript. We compare the performance of models applying adaptive clustering compared to benchmark models operated in full resolution, and we are confident that the general conclusions we can draw from such comparison can be transferred from one programming language to another.

Comment 23: BTW, I also missed the modeling time stepping; please clarify.

Reply 23: The model time stepping is 1 hour (see p. 8, line 14).

Comment 24: I would suggest the authors to clarify their methodology of the clustering. Please make more time and maybe better visualization to convey the message here. As I said earlier, I would suggest the author to allocate a full Section for the clustering method they have proposed.

Reply 23: Referee #2 also mentioned that section 2.2 (adaptive clustering) is hard to understand. We will re-write this section and the related Figures to improve comprehensibility.

Yours sincerely,

Uwe Ehret, on behalf of all co-authors

**References**

Dunger, V.: Entwicklung und Anwendung des Modells BOWAHALD zur Quantifizierung des Wasserhaushalts oberflächengesicherter Deponien und Halden. Habilitationsschrift an der Fakultät für Geowissenschaften, Geotechnik und Bergbau der TU Bergakademie Freiberg, DOI http://dx.doi.org/10.23689/fidgeo-668, 2006.

Gharari, S., M. P. Clark, N. Mizukami, W. J. M. Knoben, J. S. Wong, and A. Pietroniro (2020), Flexible vector-based spatial configurations in land models, Hydrol. Earth Syst. Sci. Discuss., 2020, 1-40.

Trigo, I. F., Dacamara, C. C., Viterbo, P., Roujean, J.-L., Olesen, F., Barroso, C., Camacho-de-Coca, F., Carrer, D., Freitas, S. C., García-Haro, J., Geiger, B., Gellens-Meulenberghs, F., Ghilain, N., Meliá, J., Pessanha, L., Siljamo, N., and Arboleda, A.: The satellite application facility for land surface analysis, International Journal of Remote Sensing, 32, 2725-2744, 10.1080/01431161003743199, 2011.

---

## Author Comment (AC2) · 6 Apr 2020

**Responses to comments posted by Referee #2**

We thank the second referee for reviewing our article and providing feedback. The referee comments identify some unclear issues and help to improve the presentation of our research. In the following, we answer to all of the comments one by one. The Referee comments are in blue. Please note that our enumeration differs from that of the referee, because we split several comments into individual points. The original order of the referee comments however remains unchanged.

Comment 1: The main objective of this paper was to propose a new way to analyze hydrological systems using adaptive clustering. The study is interesting by dynamically identifying and clustered similar model elements. Representatives per cluster inferred the dynamics. Although the application of the proposed framework looks promising, additional investigations and explanations are necessary before this paper can be published in HESS. In the next sections, I outline my major comments and suggestions that should allow the authors to improve their manuscript. Thus, my recommendation is to be accepted after the following points are clarified:

Reply 1: We are glad that the referee finds the topic of our study interesting.

Comment 2: 1. The dynamic, comparability, and similarity were emphasized in the proposed framework for adaptive clustering. However, the definitions of dynamic, comparability, and similarity are ambiguous. For example, the dynamic in hydrological simulation includes temporal dynamic and spatial dynamics. In particular, time scales for the application of hydrological models in the temporal dynamics are critical.

Reply 2: Agreed. In a revised version of the manuscript, we will provide brief definitions of relevant terms when they are first mentioned.

Comment 3: 2. The main result of high redundancy in geology and climate information is obvious. The key reason is the intricately linear or nonlinear correlation among the subsystem. The technique of adaptive clustering extracts the nonlinear correlation but not nonlinear information. Further analysis of the nonlinear relationship between subsystems is suggested.

Reply 3: We are sorry but we do not understand this comment.

Comment 4: 3. Only one basin, Attert basin, was studied. Moreover, Figure 2 and the relevant analysis did not elaborate on the geology and climate conditions, which are vital for the clustering of subsystems and hydrological simulation. The number of study areas is suggested to increase. The generalization of the framework proposed in this study is recommended to be demonstrated.

Reply 4: Spatial distribution of geology and rain gauges is indeed important for clustering. In Figure 2, geology is already indicated by the background colors (see legend 'Geology'). Rain gauge locations are also shown as yellow dots. In order to make clear their respective area of influence as calculated by the Nearest Neighbor method, we will add to the plot the related separation lines. As mentioned in the text (p8 lines 7-12), all other climatological data are taken from a single station. In addition, Table 7 shows which data are used for each sub catchment.

This paper is meant to introduce and to provide a proof-of-concept of the adaptive clustering method, and we think this can be done at the example of a single, typical hydrological model. We agree with the referee that testing the method with other models and in other catchments is clearly desirable (see our conclusion on page 29, lines 1-9), and in fact we are currently doing so. However we suggest that this is beyond the scope of this paper. In a revised version of the manuscript, we will make better clear in the abstract and in the summary and conclusions the proof-of-concept character of the paper.

Comment 5: 4. One critical issue is that the definitions of some technical terms were vague, such as aggregated characteristics and dynamics of such systems, co-evolution, catchment-uniform, and multi-criteria estimation. Please explain them in detail or replace them with easy-to-understand terms to enhance the readability of the study.

Reply 5: Will do. Please see our reply to comment 2.

Comment 6: 5. Unfortunately, the grammatical errors, confusing sentences, redundant vocabulary, and an erratic writing style hinder the message that the authors want to convey, and in some cases, render some statements ambiguous or even mistaken. I recommend that the authors encourage further to undergo a resubmission process.

Reply 6: We will re-read the paper and re-write with shorter sentences and streamlined vocabulary where necessary. Also, in the production phase, the manuscript will undergo further copy-editing.

Comment 7: Data and methods section and results section are confusing, vague wording. I suggest elaborating on the description of the adaptive clustering. Otherwise, it is hard to understand how does analyze dynamical similarity.

Reply 7: Referee #1 also mentioned that section 2.2 (adaptive clustering) is hard to understand. We will re-write this section and the related Figures to improve comprehensibility.

Comment 8: The main steps in 2.2.1 section are suggested to describe in points.

Reply 8: Please see our reply to comment 7.

Comment 9: 6. The resolution of Figure 2 is low. The information cannot be extracted.

Reply 9: Agreed. In a revised version of the manuscript, we will provide all figures in higher resolution. For a final version, all figures will be provided in separate high-resolution files as required by HESS.

Comment 10: The explanation for Figure 3 and Figure 4 is difficult to understand.

Reply 10: We will improve the explanations. Please see our reply to comment 7.

Comment 11: 7. In the data and methods section, excessive writing space is used for introducing the SHM model and its structure, which are not vital in this study. The structure of this article is suggested to adjust and enhance readability.

Reply 11: Thank you for this suggestion, which was also made by referee #1. During manuscript preparation, it was also a matter of discussion among the authors whether the SHM model description should be at such a prominent place, as it just serves as a testbed for the demonstration of adaptive clustering. Nevertheless, as SHM has not been described elsewhere so far, and as knowledge of the model structure and parameterization is important to understand its behavior in terms of dynamical similarity, we think the model should still be presented in detail. We will move most of section 2.1 (The SHM hydrological model) to the Appendix. In the main text, we will give a very brief introduction to the model and will refer to the Appendix.

Comment 12: In the results and discussion section, the principal results and conclusions are suggested to summarize briefly.

Reply 12: Respectfully we maintain that the principal results and conclusions are given, in short form, in section 4 (summary and conclusions).

Comment 13: 8. How to estimate the weights in Eq. 12? I think weight has a significant influence on the streamflow simulation in different phases, which is essential for the applicability of the proposed framework. Moreover, the Nash-Sutcliffe efficiency only prefers the simulation accuracy of high flow.

Reply 13: Assigning weights to different components of a multivariate calibration is a difficult task, and there is no single-best, objective solution for it. Rather, the choice of these weights expresses the user's subjective ranking of the components in terms of importance, and/or trustworthiness. In our case, the weights express our subjective ranking of the discharge, soil moisture, and evapotranspiration data available for calibration and validation with respect to data quality (see p. 9, line 16). In a revised version of the manuscript, we will explain this point in more detail.

NSE: We agree with the referee that NSE as a measure of model efficiency has well-known limitations (see, e.g. Schaefli and Gupta, 2007). Nevertheless, it is still one of the most widely

used performance measures in catchment hydrology, and therefore we considered it adequate for our goal of building a robust, state-of-the-art hydrological model. However, we did not only consider NSE during calibration, we also looked at mass balance errors for all water balance components (not shown in the manuscript for brevity: For discharge at the catchment outlet, it was 2.2%), and we did sequential calibration: Base flow parameters were calibrated during times of summer low flow only, interflow parameters were calibrated during times of rainfall-runoff events. We thus reduced the risk of overfitting to flood peaks, as mentioned by the referee.

Comment 14: 9. How to identify and estimate the representatives which have a strong influence on the performance of adaptive clustering.

Reply 14: For each cluster, first the number of representatives is determined by multiplying the number of catchments in the cluster with parameter 'perc_reps' (set to 10% in our study, see Table 6), which is then rounded towards an integer number. E.g. 51 catchments * 0.1 = 5.1 → 5 representatives. These 5 representatives are then randomly picked from the 51 catchments (see p. 14, lines 17-22). Random picking is a simple method, which leaves plenty of room for improvement, and we will test alternatives in the future. In a revised version of the manuscript, we will mention this is in the last section of section 4 (summary and conclusions).

Comment 15: 10. Importantly, the mechanism for the improvement of model performance was not discussed. For example, What operations lead to improvements in model performance (also involving high flow, middle flow, or low flow)?

Reply 15: We are not entirely sure we understand this question. We assume it is about how we selected the SHM Attert model structure and how we iteratively adjusted the model parameters during calibration. If this interpretation is wrong please let us know.

The model structural choice was guided by the findings of Fenicia et al. (2014) and Fenicia et al. (2016), in fact our model structure strongly resembles the models reported in them. Likewise, choosing geology as the main control of parameter variations within the catchment is based on these papers (see p. 4, lines 13-15, and p. 8, lines 1-2). Our calibration procedure is described on p. 9, lines 4-28. As explained in reply 12, our calibration strategy was sequential: Base flow parameters were calibrated during times of summer low flow only, interflow parameters were calibrated during times of rainfall-runoff events.

Yours sincerely,

Uwe Ehret, on behalf of all co-authors

**References**

Fenicia, F., D. Kavetski, H. H. G. Savenije, and L. Pfister (2016), From spatially variable streamflow to distributed hydrological models: Analysis of key modeling decisions, Water Resources Research, 52(2), 954-989.

Fenicia, F., D. Kavetski, H. H. G. Savenije, M. P. Clark, G. Schoups, L. Pfister, and J. Freer (2014), Catchment properties, function, and conceptual model representation: is there a correspondence?, Hydrol. Process., 28(4), 2451-2467.

Schaefli, B., and H. V. Gupta (2007), Do Nash values have value?, Hydrol. Process., 21(15), 2075-2080.

---

## Referee Comment (RC3) · Shervan Gharari (Referee) · 13 Apr 2020

I thank the authors for their detailed reply to my comments. For the spirit of open discussion, I would clarify/comment on some of comments/responses:

Comment 2: 2- The Introduction seems to be superficial. I would say the paper is about representation of the system in a model vs computational time/resource. In land [surface] modeling community there is significant body of literature devoted to the effect of grid size (computational burden vs spatial representation) and example of them can be Melsen et al., 2016 (and many more). This is the case in hydrological rainfall/runoff models as well (Liu et al., 2016 and many more). In its current form the Introduction

starts with general reflection on sophisticated processed-based models; then moves to concept of co-evolution (which is not directly relevant to the message this study wants to convey) and then comes down to clustering. I would suggest to re-organize the Introduction to reflect on pervious works on computational burden vs spatial representation, clustering and its application in hydrological similarities and finally make it clear what the reader should expect from this paper.

Reply 2: We thank the referee for pointing at useful literature related to the effect of grid size in land surface modeling. We will integrate it in a revised version of the manuscript. However we do not agree that the introduction is superficial. We discuss conceptual modeling and its main shortcomings to show the merits of distributed modeling. We then discuss the computational challenges of distributed modeling and how they are mitigated by adaptive time stepping adaptive gridding. We discuss the shortcomings of available adaptive gridding methods, namely that they require spatial adjacency, and argue that similarity of sub systems of natural systems is not necessarily limited to neighboring elements. Taken together, this shows that adaptive clustering i) can be useful, ii) it has novel aspects compared to existing methods of adaptive gridding, and iii) similarity is not only an artificial effect caused by representing natural systems in models, but it is also property of the real-world system (although the inevitable simplifications associated with representing real-world system in models can increase similarities). We therefore prefer to keep the structure of the introduction as it is.

Re-Reply2: What the authors mentioned here are clearer than the introduction at its current format. I would say keeping the interlocution as it is, is a disservice to the manuscript. There is only one paragraph about clustering in the introduction. For a reader, the concept of co-evolution is somehow presented as the main topic, while I think the concept of similarities of the sub-system behavior (grouped response units, GRU, hydrological response units, HRU;) should be more elaborated. The examples the authors mentioned, "north facing. . .", actually fall very well in the concept of GRU and HRU and pave the way for better presentation in the manuscript rather than co-

evolution. I agree that co-evolution is the process of creating similarities/rules but in my point of view it should not be the main point of discussion here. I leave this this to the authors and editor to decide.

Comment 6: The are kr and kb identifiable/related? it seems to a redundancy in the two processes/parameters.

Reply 6: Both parameters are indeed retention constants of linear reservoirs, however they strongly differ in magnitude. Kr is the retention constant of a single river element. As all river elements in the model are of about the same length (1 km), we used the same kr for all river elements. Its value of 1.1 hours was found by maximizing the agreement between observed and simulated streamflow in river stretches where up- and downstream gauges were available (this way we could see the effect of translation and retention in the river stretch). Kb is the retention constant of the base flow reservoir. We determined kb values, separately for each geology (see p.9, lines 4-28), by maximizing agreement of simulated and observed streamflow during times of summer low flow. Kb values are in the range of 500 hours (for Schist) and 20000 hours (for Sandstone), i.e. at least two orders of magnitude higher than kr. We are therefore confident that there is little redundancy between these processes/parameters.

Re-Reply 6: sorry I mean Ki, for the fast reservoir, instead of Kb. Indeed, the slow reservoir does not even need to be routed through river network due to its long reaction time. Ki can be said to be at the scale of hillslope temporal response which should be more or less in scale of hours. Interested to know more about Ki and how it is interpreted from the field data and separated from Kr.

Comment 9: is the satellite based evaporation a result of more sophisticated model (such as a land model)?

Reply 9: Yes, the ET estimates are produced by forcing a SVAT model (a simplified version of the ECMWF TESSEL SVAT scheme) by Land-SAF radiation products (DSSF, DSLF and AL) and ECMWF meteorology. A detailed description is given in Trigo et al. (2011), section 3.1, and on the LSA- SAF pages (https://landsaf.ipma.pt/en/products/evapotranspiration-energy-flxs/met/).

Re-Reply 9 – So good that you gave a low weight to its NS value.

Comment 13: Section 2.3.2 is again called adaptive clustering, similar to section 2.2. and again, in Section 2.3.2 the authors are referring to CATFLOW and MIKE SHE, etc.

Reply 13: Respectfully, we do not understand the concerns of the referee here.

Re-Reply 13: I meant both Sections have the same title. Also again in the middle of the manuscript it is referred to CATFLOW and other models (page 18, lines 3-4).

Comment 15: Page 20 line 9, why it is "striking" that the entropies are lower than the uniform? I would always expect so. It is also expected that the entropy is lower for the recession and higher for rising discharge. This is kind of similar to the heteroscedasticity assumption on the error as well (more diffused with higher discharges). If only observation is used with varying error assumption, higher streamflow will have higher entropy and lower streamflow will have lower entropy.

Reply 15: We agree that it is no surprise to see entropies below the entropy of the corresponding uniform distribution. 'Striking' here refers to the fact that entropies are well below the uniform entropy, and often close to zero. In our opinion this is indeed noteworthy, and it shows the high potential for adaptive clustering. In a revised version of the manuscript, we will make this point clear. We also agree with the referee that hydrologists have known since long that the degree of similarity between sub systems varies with the hydrological situation. The point we want to make here is that i) entropy of normalized, binned distributions of states and fluxes expresses this in a conveniently dimensionless way, and ii) that we make use of this knowledge.

Re-Reply 15: I would say it is not still sticking. In my point of view any model set up (even worst ones) can easily show very good behaving entropy as they are mostly affected by forcing and memory of the forcing rather than parameters.

Comment 19: 6- For me personally, moving from the world of conceptual models to land models, I would like to question the motivation of this study. Although saving time is valuable but having method that needs model re-run or updating for more complex model is terribly cumbersome. This is the reason why the authors have chosen to use SHM rather than CATFLOW for example.

Reply 19: The referee correctly states one main motivation for this study: saving computation time. This is already a more than sufficient reason, as the referee will surely agree, as he also works on concepts to make high-resolution land surface modeling more efficient (Gharari et al, 2020). In addition, the concept of adaptive clustering yields valuable insights in the time-and space patterns of similarity among sub systems, which, we daresay, is a useful contribution to hydrology research. We have chosen SHM for the proof-of-concept as any hydrologist can easily connect to it, and hence we can show the effects of adaptive clustering in an environment familiar to most hydrologists. We agree with the referee that implementing adaptive clustering in more advanced models will be more demanding, but also the gains will be higher (see p. 29, lines 5-9). So it will be well worth the try.

Re-Reply 19: Yes, testing the method for the more sophisticated models is desirable of course. I just wanted to draw the attention of the authors that to the fact that running a more complex model means more technicalities. Given those technicalities, and time/resources to fix them, it is not really clear if the final gain will be higher. The technicalities can be how to efficiently read/write/update this adoptive clustering; how to efficiently do a warm start for a model; how to pass this over various processors if needed; do the mentioned models' capabilities allow such an approach? Etc.

I give an analogy of the sensitivity analysis of land models. Land models may fail (crash) for some given parameter sets therefore may not result in output values (objective functions) which are essential given the struct of parameter sampling method. This may cause issues for sensitivity methods which should be thought through. I would say adding one or two sentences on these technicalities/obstacles at the end might be useful

for the reader.

With regards,

Shervan Gharari

---

## Author Comment (AC3) · 23 Apr 2020

**Responses to RC 3 posted by Referee #1**

We thank the first referee, Shervan Gharari, for the interest in our study and the replies to our questions. In the following, we answer to all of the comments one by one. The Referee comments are in blue.

Comment 2: 2- The Introduction seems to be superficial. […] I would say the paper is about representation of the system in a model vs computational time/resource. In land [surface] modeling community there is significant body of literature devoted to the effect of grid size (computational burden vs spatial representation) and example of them can be Melsen et al., 2016 (and many more). This is the case in hydrological rainfall/runoff models as well (Liu et al., 2016 and many more). In its current form the Introduction starts with general reflection on sophisticated processed-based models; then moves to concept of co-evolution (which is not directly relevant to the message this study wants to convey) and then comes down to clustering. I would suggest to re-organize the Introduction to reflect on pervious works on computational burden vs spatial representation, clustering and its application in hydrological similarities and finally make it clear what the reader should expect from this paper.

Reply 2: We thank the referee for pointing at useful literature related to the effect of grid size in land surface modeling. […] We will integrate it in a revised version of the manuscript. However we do not agree that the introduction is superficial. We discuss conceptual modeling and its main shortcomings to show the merits of distributed modeling. We then discuss the computational challenges of distributed modeling and how they are mitigated by adaptive time stepping and adaptive gridding. We discuss the shortcomings of available adaptive gridding methods, namely that they require spatial adjacency, and argue that similarity of sub systems of natural systems is not necessarily limited to neighboring elements. Taken together, this shows that adaptive clustering i) can be useful, ii) it has novel aspects compared to existing methods of adaptive gridding, and iii) similarity is not only an artificial effect caused by representing natural systems in models, but it is also property of the real-world system (although the inevitable simplifications associated with representing real-world system in models can increase similarities). We therefore prefer to keep the structure of the introduction as it is.

Re-Reply 2: What the authors mentioned here are clearer than the introduction at its current format. I would say keeping the interlocution as it is, is a disservice to the manuscript. There is only one paragraph about clustering in the introduction. For a reader, the concept of co-evolution is somehow presented as the main topic, while I think the concept of similarities of the sub-system behavior (grouped response units, GRU, hydrological response units, HRU;) should be more elaborated. The examples the authors mentioned, "north facing. . .", actually fall very well in the concept of GRU and HRU and pave the way for better presentation in the manuscript rather than co- evolution. I agree that co-evolution is the process of creating

similarities/rules but in my point of view it should not be the main point of discussion here. I leave this this to the authors and editor to decide.

Re-Re-Reply 2: We agree that the concept of GRU/HRUs, namely time-invariant grouping of model elements without the requirement of spatial adjacency is not discussed in the introduction, but relevant to the topic. We will add a related discussion to the introduction in a revised version of the manuscript. Nevertheless, we prefer keeping the part about co-evolution, as it explains why the similarities that we exploit by GRU/HRU or adaptive clustering methods occur.

Comment 6: The are kr and kb identifiable/related? it seems to a redundancy in the two processes/parameters.

Reply 6: Both parameters are indeed retention constants of linear reservoirs, however they strongly differ in magnitude. Kr is the retention constant of a single river element. As all river elements in the model are of about the same length (1 km), we used the same kr for all river elements. Its value of 1.1 hours was found by maximizing the agreement between observed and simulated streamflow in river stretches where up- and downstream gauges were available (this way we could see the effect of translation and retention in the river stretch). Kb is the retention constant of the base flow reservoir. We determined kb values, separately for each geology (see p.9, lines 4-28), by maximizing agreement of simulated and observed streamflow during times of summer low flow. Kb values are in the range of 500 hours (for Schist) and 20000 hours (for Sandstone), i.e. at least two orders of magnitude higher than kr. We are therefore confident that there is little redundancy between these processes/parameters.

Re-Reply 6: sorry I mean Ki, for the fast reservoir, instead of Kb. Indeed, the slow reservoir does not even need to be routed through river network due to its long reaction time. Ki can be said to be at the scale of hillslope temporal response which should be more or less in scale of hours. Interested to know more about Ki and how it is interpreted from the field data and separated from Kr.

Re-Re-Reply 6: The time scale of river routing is indeed negligible compared to that of the base flow, so from a practical point there is not much of a difference whether base flow is routed or not. But as it actually enters the river before the catchment outlet (otherwise the rivers would fall dry in summer, which they did not even in the very dry summer 2015), we routed it. Kr we found, as mentioned in our initial reply, by maximizing the agreement between observed and simulated streamflow in river stretches where up- and downstream gauges were available. The values of Ki were determined by calibration. As mentioned in the manuscript, for the entire model setup and choice of reasonable parameter ranges, we relied on the detailed investigations and findings of Fenicia et al. (2014, 2016).

Comment 9: is the satellite based evaporation a result of more sophisticated model (such as a land model)?

Reply 9: Yes, the ET estimates are produced by forcing a SVAT model (a simplified version of the ECMWF TESSEL SVAT scheme) by Land-SAF radiation products (DSSF, DSLF and AL) and ECMWF meteorology. A detailed description is given in Trigo et al. (2011), section 3.1, and on the LSA-SAF pages (https://landsaf.ipma.pt/en/products/evapotranspiration-energy-flxs/met/).

Re-Reply 9:  – So good that you gave a low weight to its NS value.

Re-Re-Reply 9: Indeed this ETP product is not a rock-hard observed truth, but nevertheless it was a useful benchmark to evaluate our model in terms of daily and seasonal ETP patterns both qualitatively and quantitatively.

Comment 13: Section 2.3.2 is again called adaptive clustering, similar to section 2.2. and again, in Section 2.3.2 the authors are referring to CATFLOW and MIKE SHE, etc.

Reply 13: Respectfully, we do not understand the concerns of the referee here.

Re-Reply 13: I meant both Sections have the same title. Also again in the middle of the manuscript it is referred to CATFLOW and other models (page 18, lines 3-4).

Re-Re-Reply 13: We will change the titles in a revised version of the manuscript. But we still do not understand why we should not refer to examples of physically based hydrological models in this context.

Comment 15: Page 20 line 9, why it is "striking" that the entropies are lower than the uniform? I would always expect so. It is also expected that the entropy is lower for the recession and higher for rising discharge. This is kind of similar to the heteroscedasticity assumption on the error as well (more diffused with higher discharges). If only observation is used with varying error assumption, higher streamflow will have higher entropy and lower streamflow will have lower entropy.

Reply 15: We agree that it is no surprise to see entropies below the entropy of the corresponding uniform distribution. 'Striking' here refers to the fact that entropies are well below the uniform entropy, and often close to zero. In our opinion this is indeed noteworthy, and it shows the high potential for adaptive clustering. In a revised version of the manuscript, we will make this point clear. We also agree with the referee that hydrologists have known since

long that the degree of similarity between sub systems varies with the hydrological situation. The point we want to make here is that i) entropy of normalized, binned distributions of states and fluxes expresses this in a conveniently dimensionless way, and ii) that we make use of this knowledge.

Re-Reply 15: I would say it is not still sticking. In my point of view any model set up (even worst ones) can easily show very good behaving entropy as they are mostly affected by forcing and memory of the forcing rather than parameters.

Re-Re-Reply 15: We would like to reply along two aspects the referee mentions: The first is about forcing as a control of similarity, the other is about a high degree of similarity observed in many models.

Forcing: We agree that forcing is an important control of sub catchment dynamical similarity, but it is not the only one. Just one example: Two hillslopes may behave identical during snowmelt conditions if they are exposed to radiation in the same manner. But they may function quite differently in summer if one of them is vegetated and the other is not.

Similarity: We agree that most distributed models show a high degree of similarity among the states and fluxes of its sub elements, which is exactly why methods exploiting this similarity such as adaptive gridding, GRUs or HRUs are so useful. However, by far not all models make use of these methods. We therefore think it is noteworthy to discuss similariy, and especially the time-variant character of similarity, to show the potential computational savings.

Comment 19: 6- For me personally, moving from the world of conceptual models to land models, I would like to question the motivation of this study. Although saving time is valuable but having method that needs model re-run or updating for more complex model is terribly cumbersome. This is the reason why the authors have chosen to use SHM rather than CATFLOW for example.

Reply 19: The referee correctly states one main motivation for this study: saving computation time. This is already a more than sufficient reason, as the referee will surely agree, as he also works on concepts to make high-resolution land surface modeling more efficient (Gharari et al, 2020). In addition, the concept of adaptive clustering yields valuable insights in the time-and space patterns of similarity among sub systems, which, we daresay, is a useful contribution to hydrology research. We have chosen SHM for the proof-of-concept as any hydrologist can easily connect to it, and hence we can show the effects of adaptive clustering in an environment familiar to most hydrologists. We agree with the referee that implementing adaptive clustering in more advanced models will be more demanding, but also the gains will be higher (see p. 29, lines 5-9). So it will be well worth the try.

Re-Reply 19: Yes, testing the method for the more sophisticated models is desirable of course. I just wanted to draw the attention of the authors that to the fact that running a more complex model means more technicalities. Given those technicalities, and time/resources to fix them, it is not really clear if the final gain will be higher. The technicalities can be how to efficiently read/write/update this adoptive clustering; how to efficiently do a warm start for a model; how to pass this over various processors if needed; do the mentioned models' capabilities allow such an approach? Etc.

I give an analogy of the sensitivity analysis of land models. Land models may fail (crash) for some given parameter sets therefore may not result in output values (objective functions) which are essential given the struct of parameter sampling method. This may cause issues for sensitivity methods which should be thought through. I would say adding one or two sentences on those technicalities/obstacles at the end might be useful for the reader.

Re-Re-Reply 19: We agree with the referee that it remains to be proven that the benefits of adaptive clustering will not be eaten up by its overhead in more sophisticated applications. And in fact it would be highly interesting to discuss with the referee the potential of adaptive clustering in the land surface models he uses. We are carefully optimistic about the potential benefits of adaptive clustering, as established methods to save computation time, like adaptive gridding and adaptive time stepping, also come with a considerably overhead and are nevertheless very effective. Nevertheless, in a revised version of the manuscript, we will add to the conclusions a short discussion about potential challenges when applying the method to more sophisticated models.

Yours sincerely,

Uwe Ehret, on behalf of all co-authors

**References**

Fenicia, F., Kavetski, D., Savenije, H. H. G., and Pfister, L.: From spatially variable streamflow to distributed hydrological models: Analysis of key modeling decisions, Water Resources Research, 52, 954-989, 10.1002/2015wr017398, 2016.

Fenicia, F., Kavetski, D., Savenije, H. H. G., Clark, M. P., Schoups, G., Pfister, L., and Freer, J.: Catchment properties, function, and conceptual model representation: is there a correspondence?, Hydrol. Process., 28, 2451-2467, 10.1002/hyp.9726, 2014.

---

## Author Response (AR1)

**Cover letter to revision of hess-2020-65**

Dear Editor, dear Referees,

We have completed a comprehensive revision of the manuscript – in fact we changed much more than announced in the replies to the referees. We hope the revised version is not just different from the previous, but conveys our message better. Before going into the details of the point-by-point replies to the referees, we would like to mention the main changes. Please note that due to the large amount of text changed, we refrained from providing a manuscript version with all changes tracked. Instead we highlight major changes by red text: If an entire section was largely re-written, we highlight the section header in red. Changes of particular words, sentences or passages are also highlighted in red.

**Main changes**

1.  We have changed the title
2.  The general focus of the manuscript is now on suggesting adaptive clustering as a tool for making distributed modelling more efficient. Using adaptive clustering as a tool for hydrological system analysis is still discussed, but not as prominently as before.
3.  Throughout the text, we make better clear that the presented study serves as a proof-of-concept
4.  We provide definitions of the most important terms in the introduction: Similarity, redundancy, time-invariant, static, time-variant, dynamic, adaptive.
5.  We have checked the entire article for grammar and spelling. We replaced long sentences by shorter ones.
6.  We have re-written the abstract
7.  We have re-written most of the introduction and added a discussion of the HRU, REA, REW and GRU concepts. We also added a discussion about the effect of grid size and computational expenses.
8.  We have re-written the explanation of the adaptive clustering method (new section 2.1) in a step-by-step manner.
9.  We have shifted most of the sections about the SHM model to the new Appendices A1-A3.
10. We have re-written the implementation of adaptive clustering in a step-by-step manner (new section 2.3)
11. At the beginning of new section 2.4, and at the beginning of section 3, we establish a connection between similarity analysis (2.4.1 and 3.1) and adaptive clustering (2.4.2 and 2.3).
12. We have added short summaries at the end of sections 3.1 and 3.2
13. We have added to the summary and conclusion a paragraph about further work, and challenges of integrating adaptive clustering into more sophisticated models.

**Point-by-point replies to referee comments**

Please note that we provide page/line numbers and section numbers for the new manuscript version.

**RC 1 and RC 3 (Shervan Gharari)**

Comment 1: Please see point 5 in 'main changes. Also, the manuscript will undergo further copy-editing in the production process.

Comment 2: Please see point 7 in 'main changes'.

Comment 3: Please see point 9 in 'main changes'. In section 2, we now first provide the explanation of the adaptive clustering method (section 2.1), and then a brief introduction to the study area and the hydrological model (section 2.2), referring to the appendices where appropriate.

Comment 4: In the revised version of the manuscript, we replaced 'crop coefficient' by 'vegetation correction factor' in Table A1.

Comment 5: In the revised version of the manuscript, we mention that there are 147 river elements (p 22 line 8)

Comment 6: No changes made. Please see our replies to this comment from the discussion phase.

Comment 7: In Appendix A.2, explaining the Attert model setup, we now refer to Table 2 with the list of sub catchment geology and land use classes. We also added to Fig. 2 separation lines indicating the area of influence of each rain gauge as determined by the Nearest Neighbour method, and refer to it from Appendix A.2 (p 22 line 12).

Comment 8: No changes made. Please see our replies to this comment from the discussion phase.

Comment 9: No changes made. Please see our replies to this comment from the discussion phase.

Comment 10: We have moved the detailed explanation of the SHM structure and processes to Appendix A1. The description of set up, calibration and validation of the SHM model to the Attert basin is now in appendices A.2 and A.3. In section 2, we now first provide the explanation of the adaptive clustering method (section 2.1), and then a brief introduction to the study area and the hydrological model (section 2.2), referring to the appendices where appropriate.

Comment 11: We have changed most sub section titles. Sub section 2.2.1 is removed in the revised version.

Comment 12: Section 2.3.1 from the previous version of the manuscript is now section 2.4.1 titled 'Entropy as a measure of hydrological similarity'.

Comment 13: We have changed most sub section titles. Old section 2.2 is now section 2.1, titled 'Adaptive clustering'. Old section 2.3.2 is now section 2.4.2 titled 'Evaluation criteria and benchmark models for adaptive clustering'

Comment 14: Please see points 2 and 11 in 'main changes'.

Comment 15: In the revised version of the manuscript, we rephrased the text about the entropy results (p. 11 line 9 pp)

Comment 16: We have added a brief summary of the main findings of section 3.1 at the section end, and make a connection to the next section.

Comment 17: Please see our reply to comment 16 above, and our replies to this comment from the discussion phase.

Comment 18: We have completely re-written the abstract, and substantially revised the summary & conclusion section.

Comment 19: We have added at the end of the summary & conclusions a brief discussion about potential problems of including adaptive clustering in more advanced models.

Comment 20: No changes made. Please see our replies to this comment from the discussion phase.

Comment 21: No changes made. Please see our replies to this comment from the discussion phase.

Comment 22: We have added a few words to the abstract and the summary and conclusion stating that the main purpose of the paper is a proof-of-concept of adaptive clustering, and therefore the choice of the programming environment and the computational performance is not of major importance.

Comment 23: No changes made. Please see our replies to this comment from the discussion phase.

Comment 24: Please see point 8 in 'main changes'.

**RC 2 (Anonymous)**

Comment 1: No changes made. Please see our reply to this comment from the discussion phase.

Comment 2: Please see point 4 in 'main changes'.

Comment 3: No changes made. Please see our previous reply to this comment

Comment 4:

- We added to Fig. 2 separation lines indicating the area of influence of each rain gauge as determined by the Nearest Neighbour method.
- We completely re-wrote the abstract, and partly the summary and conclusion, adding comments that the main purpose of the paper is a proof-of-concept of adaptive clustering.

Comment 5: Please see point 4 in 'main changes'.

Comment 6: Please see point 5 in 'main changes. Also, the manuscript will undergo further copy-editing in the production process.

Comment 7: Please see points 8-10 in 'main changes'.

Comment 8: Please see points 8-10 in 'main changes'.

Comment 9: All figures in the manuscript are now in higher resolution.

Comment 10: We have replaced old Fig. 3 with a new one (now Fig. 1), and also changed the related explanation. We have removed old Fig. 4.

Comment 11: Please see point 9 in 'main changes'.

Comment 12: Please see point 12 in 'main changes'.

Comment 13: We have added to the manuscript (p 23 lines 5-9) a brief explanation about our choice of the weights for each component of the objective function.

Comment 14: We have added at the end of the summary & conclusions a sentence about how the choice of representatives can be improved in future versions of adaptive clustering

Comment 15: No changes made. Please see our reply to this comment from the discussion phase.

Yours sincerely,

Uwe Ehret, on behalf of all co-authors

---

## Author Response (AR2)

**Responses to review of revised manuscript by Referee #2**

Dear Editor,

Please find below our point-by-point replies to the referee comments on the revised version of our manuscript. The Referee comments are in blue. The page and line numbers refer to the revised version of the manuscript.

Comment 1: As fresh read of the revised paper, I think the paper reads well for most of aspects: paper structure, descriptions, focus, figures. Reviewing the responses, the authors addressed my comments adequately. I appreciate the revisions made based on my comments, but I still feel that the statement for adaptive clustering could be improved (though I do see the effort to revise that section). I think one minor revision would improve the paper.

Reply 1: We are sorry, but we do not understand where the referee misses a stronger statement for adaptive clustering. We think we made its key goals clear in the abstract (page 1 line 10) and in the first sentence of the summary and conclusion (page 18 line 6).

Comment 2: 1. There are different properties between the state variables and flux variables in models. How to distinguish and handle them in the system of adaptive clustering?

Reply 2: The mapping from representatives to recipients within a cluster is done in the same manner for state and flux variables: Each normalized state (or flux) from the representative is transferred to the recipient, and then de-normalized by the particular [min,max] range of the recipient for the particular state (or flux). We explain this in section 2.1, but mainly only mentioned state variables. For clarification, we have added the term 'flux' here (step 'e' on page 4 lines 9-10). In section 2.3, where we explain the implementation of adaptive clustering in the SHM Attert model, we already mentioned both state and flux variables (page 7 lines 18-23).

Comment 3: 2. The similarity of spatial patterns and their clustering control is elaborated; however, the statement on the similarity of temporal patterns is not clear. Moreover, how to solve the transition between two time series with different states and fluxes of all sub catchments while the model run?

Reply 3: We assume the referee refers to page 13, line 16, and page 18, line 22 here. What we mean by temporal patterns of similarity is that the degree of similarity among the sub catchments varies with time (i.e. the number of clusters needed varies over time). Also, which sub catchments are put into one cluster also varies with time: Sometimes, when rainfall dominates, the sub catchments close to the same rain gauge fall into one group. At other times, sub catchments sharing the same geology will be put into one group. To clarify this point, we replaced at the above-mentioned places "the temporal and spatial patterns of similarity" by "the spatial patterns of similarity and their variation with time".

Comment 4: 3. In Figure 3 (a), the colors cannot be distinguished.

Reply 4: We changed the color of $q_{out}$. Now all lines should be distinguishable.

Comment 5: 4. The conclusions and summary are suggested to condensed, and the key points are focused.

Reply 5: We expanded this section according to the recommendations of referee#1, so we prefer to keep it as is.

Yours sincerely,

Uwe Ehret, on behalf of all co-authors